



**At which time scale does the complementary principle perform best on evaporation estimation?**

Liming Wang[1], Songjun Han[2], Fuqiang Tian[1*].

[1] Department of Hydraulic Engineering, State Key Laboratory of Hydroscience and

Engineering, Tsinghua University, Beijing 100084, China

[2] State Key Laboratory of Simulation and Regulation of Water Cycle in River Basin,

China Institute of Water Resources and Hydropower Research, Beijing 100038, China

**Correspondence to:**

Fuqiang Tian: tianfq@mail.tsinghua.edu.cn





**Abstract**
The complementary principle has been widely used to estimate evaporation under different
conditions. However, it remains unclear that at which time scale the complementary principle
performs best. In this study, evaporation estimation was assessed over 88 eddy covariance
(EC) monitoring sites at multiple time scales (daily, weekly, monthly, and yearly) by using
the sigmoid and polynomial generalized complementary functions. The results indicate that
the generalized complementary functions exhibit the highest skill in estimating evaporation at
the monthly scale. The uncertainty analysis shows that this conclusion is not affected by
ecosystem types nor energy correction methods. Through comparisons at multiple time
scales, we found that the slight difference between the two generalized complementary
functions only exists when the independent variable ($x$) in the functions approaches 1. The
difference results in different performance of the two models at daily and weekly scales.
However, such difference vanishes at monthly and annual time scales as few high $x$
occurrences. This study demonstrates the applicability of the generalized complementary
functions across multiple time scales and provides a reference for choosing the suitable
timestep for evaporation estimation in relevant studies.



**Keywords:**
Evaporation; Generalized complementary functions; Multiple time scales; Ecosystem types;
Energy correction methods



## 1. Introduction

Terrestrial evaporation ($E$) including soil evaporation, wet canopy evaporation, and plant

transpiration, is one of the most important components in global water and energy cycles

(Wang and Dickinson, 2012). The evaporation process affects the atmosphere by a series of

feedbacks on humidity, temperature, and momentum (Brubaker and Entekhabi, 1996; Neelin

et al., 1987; Shukla and Mintz, 1982). Quantifying evaporation is crucial for a deep

understanding of water and energy interactions between the land surface and the atmosphere.

Generally, the meteorological studies focus on the evaporation change at hourly and daily

scales; the hydrological applications need evaporation data at weekly, monthly or longer time

scales (Morton, 1983); and the climate change researches pay more attention to the

interannual variation. The observation of $E$ can be operated at different time scales. For

example, the Eddy covariance, lysimeter, and scintillometer can measure the evaporation at

the half-hour scale, and the water balance methods can observe the evaporation at monthly to

yearly scales (Wang and Dickinson, 2012). However, in most situations the observation is

unavailable and the estimation of $E$ is necessary. There are several types of methods for

evaporation estimation, for example, the Budyko-type methods (Budyko, 1974; Fu, 1981),

the Penman-type methods (Penman, 1948; Monteith, 1965) and the complementary-type

methods (Bouchet, 1963; Brutsaert and Stricker, 1979). The Budyko-type methods perform

well at annual or longer time scales; the Penman-type methods can be applied at hourly and

daily scales; while the complementary-type methods are used at multiple time scales (Crago

and Crowley, 2005; Han and Tian, 2018; Crago and Crowley, 2018; Ma et al., 2019) without

an explicit cognization of the time scale issue.





Recently, the complementary principle, as one of the major types of $E$ estimation methods,
has drawn increasing attention because it can be implemented with standard meteorological
data (radiation, wind speed, air temperature, and humidity) without the requirement for
complicated underlying surface properties. Based on the coupling between the land surface
and the atmosphere, the complementary principle assumes that the limitation of the wetness
state in the underlying surface on evaporation can be synthetically reflected by the
atmospheric wetness (Han et al., 2020). Bouchet (1963) first proposed the "complementary
relationship" (CR), which suggested that the apparent potential evaporation ($E_{pa}$) and the
actual $E$ depart from potential evaporation ($E_{po}$) in equal absolute values but opposite
directions ($E_{pa} - E_{po} = E_{po} - E$). Subsequently, the CR was extended to a linear function with
an asymmetric parameter (Brutsaert and Parlange, 1998). Further studies found that the linear
function underestimates $E$ in arid environments and overestimates $E$ in wet environments
(Han et al., 2008; Hobbins et al., 2001; Qualls and Gultekin, 1997). To address the issue, Han
et al. (2011; 2012; 2018) proposed a sigmoid generalized complementary function (SGC, see
equation (1) for detail). As a modification to the AA approach, the SGC function illustrates
the relationship between two dimensionless terms, $E/E_{pen}$ and $E_{rad}/E_{pen}$, where $E_{pen}$ is the
Penman evaporation (Penman, 1948) and $E_{rad}$ is the radiation term of $E_{pen}$. The SGC function
shows higher accuracy in estimating $E$ (Han and Tian, 2018; Ma et al., 2015b; Zhou et al.,
2020) and outperforms the linear functions, especially in dry desert regions and wet
farmlands (Han et al., 2012). Obtaining the impetus from Han et al. (2012), Brutsaert (2015)
proposed a quartic polynomial generalized complementary function (PGC, see equation (5)
for detail). The PGC function describes the relationship between $E/E_{pa}$ and $E_{po}/E_{pa}$, where $E_{pa}$



and $E_{po}$ are formulated in the manner of the AA approach. The PGC function has also been
frequently used in recent years (Brutsaert et al., 2017; Hu et al., 2018; Liu et al., 2016; Zhang
et al., 2017).

The prerequisite of the complementary principle is the adequate feedback between the land
surface and the atmosphere, which results in an equilibrium state. In this situation, the
wetness condition of the land surface can be largely represented by the atmospheric
conditions. Therefore, the time scales used in the complementary principle need to satisfy the
adequate feedback assumption. However, this issue involves the complex processes of
atmospheric horizontal and vertical motion, and it is difficult to be explained theoretically.
Morton (1983) noticed this problem earlier and suggested that the complementary principle is
not suitable for short time scale (e.g., less than 3 days) mainly because of the potential lag
times associated with the response of energy and water vapor storage to disturbances in the
atmospheric boundary layer. However, there is no solid evidence or theoretical identification
to support this inference. The original complementary relationship and the AA function are
not limited by the applicable time scales. In the derivation of the advanced generalized
complementary functions (SGC of Han and Tian (2018) and PGC of Brutsaert (2015)), no
specific time scale is defined neither. In practice, the complementary principle has been
widely adopted to estimate $E$ at multiple time scales including hourly (Crago and Crowley,
2005; Parlange and Katul, 1992), daily (Han and Tian, 2018; Ma et al, 2015b), monthly (Ma
et al, 2019; Brutsaert, 2019), and annual scales (Hobbins et al., 2004). The accuracy of the
results varied in different studies. Crago and Crowley (2005) found the linear complementary





function performs well in estimating $E$ at small time scales less than half-hour using the data
from several famous experimental projects (e.g., International Satellite Land Surface
Climatology Project). The correlation coefficient between simulated $E$ and observed $E$ ranges
from 0.87 to 0.92 in different experiments. The results of Ma et al. (2015b) indicated that the
SGC function (RMSE = 0.39 mm day$^{-1}$) is competent in estimating $E$ in an alpine steppe
region of the Tibetan Plateau at the daily scale. Han and Tian (2018) applied the SGC
function on the daily data of 20 EC sites from the FLUXNET and found it performs well in
estimating $E$ with a mean Nash-Sutcliffe efficiency (NSE) value of 0.66. Crago and Qualls
(2018) evaluated the PGC function and their rescaled complementary functions using the
weekly data of 7 FLUXNET sites in Australia, and the results showed that all the functions
perform adequately with a correlation coefficient between simulated $E$ and observed $E$ higher
than 0.9. Ma et al. (2019) also validated an emendatory polynomial complementary function
at the monthly scale, and the NSE values of 13 EC sites in China are higher than 0.72. At the
annual scale, Zhou et al. (2020) found the mean NSE of the SGC function is 0.28 for 15
catchments in the Loess Plateau. Since these results were derived with different functions
under varied conditions, it is difficult to determine at which time scale the performance is the
best, and it is more difficult to explain theoretically how long the land-atmosphere feedback
needs to achieve equilibrium.

In previous studies, the model validations were mostly completed at daily scale (Brutsaert,
2017; Han and Tian 2018; Wang et al. 2020), and the datasets of evaporation estimation were
often established at monthly scale (Ma et al., 2019; Brutsaert et al., 2019). However, each



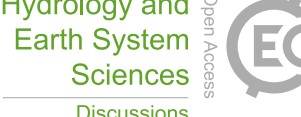

study only focused on a single timescale. In this study, we assessed the performance of the
complementary functions on evaporation estimation at multiple time scales (daily, weekly,
monthly, and yearly). The assessment was carried out over 88 EC monitoring sites with > 5-
year-long observation records. In view of the fact that the complementary principle has
developed to the nonlinear generalized forms, we selected two nonlinear complementary
functions in the literature, i.e., the SGC function (Han et al., 2012; 2018) and the PGC
function (Brutsaert, 2015). The key parameters of the complementary functions need to be
determined by calibration. We chose the uniform database and the uniform parameter
calibration method for the optimization of the two complementary functions. We aimed to
determine the most suitable timescale for the complementary functions through comparison
of the performances at different timescales. It's important not only for the deep understanding
of the application of the complementary principle, but also for the timestep selection in the
evaporation database establishment and evaporation trend analysis.

This paper is organized as follows. Section 1 briefly describes the development of the
complementary theory and our motivations to investigate the timescale issue. Section 2
describes the two functions, the parameter calibration method, and the data sources and
processing. Section 3 shows and discusses the performance of the complementary functions
at multiple time scales, the dependence of the key parameters on time scales, and the
uncertainties in the analysis. The conclusions are given in Section 4.

**2. Methodology**



**2.1 The sigmoid generalized complementary function**
Han et al. (2012; 2018) proposed a generalized form of the complementary function that
expresses $E/E_{pen}$ as a sigmoid function (SGC) of $E_{rad}/E_{pen}$:
$$y = \frac{E}{E_{Pen}} = \frac{1}{1 + m\left(\frac{x_{max} - x}{x - x_{min}}\right)^n}$$
$$x = \frac{E_{rad}}{E_{Pen}} \tag{1}$$
where $x_{max}$ corresponds to the certain maximum value of $x$ under extremely wet
environments, and $x_{min}$ corresponds to the certain minimum value of $x$ under extremely arid
environments. In this study, $x_{max}$ and $x_{min}$ were set as 1 and 0, respectively, for convenience.
The $E_{pen}$ term is defined by Penman's equation (Penman, 1950; Penman, 1948), which can be
expressed as
$$E_{pen} = \frac{\Delta(R_n - G)}{\Delta + \gamma} + \frac{\rho c_p}{\Delta + \gamma} \frac{\kappa^2 u}{ln(\frac{z-d_0}{z_{0m}}) ln(\frac{z-d_0}{z_{0v}})} (e_a^* - e_a) \tag{2}$$
where, $\Delta$ (kPa C$^{-1}$) is the slope of the saturation vapor curve at air temperature; $R_n$ is the net
radiation; $G$ is the ground heat flux; $\gamma$ (kPa C$^{-1}$) is a psychrometric constant; $\rho$ is the air
density; $c_p$ is the specific heat; $\kappa = 0.4$ is the von Karman constant; $u$ is the wind speed at
measurement height; $e_a^*$ and $e_a$ are the saturated and actual vapor pressures of air,
respectively; $z$ is the measurement height (Table S1); $d_0$ is the displacement height; $z_{0m}$ and
$z_{0v}$ are the roughness lengths for momentum and water vapor, respectively, which are
estimated from the canopy height ($h_c$, Table S1), $d_0 = 0.67h_c$, $z_{0m} = 0.123h_c$, and $z_{0v} =$
$0.1z_{0m}$ (Monin and Obukhov, 1954; Allen et al., 1998). $E_{rad}$ is the radiation term of the
Penman evaporation:
$$E_{rad} = \frac{\Delta(R_n - G)}{\Delta + \gamma} \tag{3}$$





The two parameters $m$ and $n$ of equation (1) can be determined by the Priestley-Taylor
coefficient $\alpha$ and the asymmetric parameter $b$ (Han and Tian, 2018).
$$\begin{cases} n = 4\alpha(1 + b^{-1})x_{0.5}(1 - x_{0.5}) \\ m = (\frac{x_{0.5}}{1-x_{0.5}})^n \end{cases} \tag{4}$$

where, $x_{0.5}$ is a variable that corresponds to $y = 0.5$, and equals to $\frac{0.5+b^{-1}}{\alpha(1+b^{-1})}$.

**2.2 The polynomial generalized complementary function**
Brutsaert (2015) proposed the polynomial generalized complementary (PGC) function, which
describes the relationship between $E/E_{pa}$ and $E_{po}/E_{pa}$. According to the AA approach
(Brutsaert and Stricker, 1979), $E_{pa}$ is formulated by Penman's (1948) equation ($E_{pen}$), and $E_{po}$
is formulated by Priestley-Taylor's (1972) equation ($E_{PT} = \alpha\,E_{rad}$). We uniformed the
independent variable as $E_{rad}/E_{pen}$ to compare the two functions conveniently, and the
polynomial function can be expressed as:
$$y = (2 - c)\alpha^2 x^2 - (1 - 2c)\alpha^3 x^3 - c\alpha^4 x^4 \tag{5}$$

where, $c$ is an adjustable parameter. When $c = 0$, equation (5) reduce to
$$y = 2\alpha^2 x^2 - \alpha^3 x^3 \tag{6}$$


**2.3 Parameter optimization method**
In this study, $\alpha$ was calculated by the mean value of $E/E_{rad}$ whenever $E/E_{pen}$ is larger than 0.9
(Kahler and Brutsaert, 2006; Ma et al., 2015a). When all the $E/E_{pen}$ values are less than 0.9, $\alpha$
was set as the default value of 1.26. The key parameter $b$ in SGC was calibrated by an
optimization algorithm with the objective function as minimization the mean absolute error
(MAE) between the estimated $E$ (by equation (1)) and the observed $E$. Similarly, the key



parameter $c$ in PGC was calibrated by an optimization algorithm with the objective function
as minimization the MAE between the estimated $E$ (by equation (5)) and the observed $E$.

**2.4 Data sources and data processing**
The eddy flux data analyzed in this study were obtained from the FLUXNET database
(http://fluxnet.fluxdata.org, Baldocchi et al., 2001). Observations from a total of 88 sites
around the world were analyzed. The detailed information on these sites is listed in Table S1.
These sites were selected from the FLUXNET database because they have observations for
longer than 5 years. The 88 sites include 11 IGBP (International Geosphere-Biosphere
Programme) land cover classes: ENF, evergreen needleleaf forests (27 sites); EBF, evergreen
broadleaf forests (8); DBF, deciduous broadleaf forests (13); MF, mixed forests (5); OSH,
open shrublands (4); CSH, closed shrublands (1); WSA, woody savannas (3); SAV, savannas
(4); GRA, grasslands (15); CRO, croplands (6); WET, permanent wetlands (2). The climate
of the 88 sites ranges from arid to humid. Among the 88 sites, 11 sites have mean annual
precipitation lower than 200 mm, 47 sites have precipitation between 200 ~ 500 mm and 30
sites have precipitation above 500 mm. Eleven sites are located in the Southern Hemisphere
(i.e., Australia, Brazil, and South Africa) and the others are located in the Northern
Hemisphere.

Variables including net radiation, sensible heat flux, latent heat flux, ground heat flux, wind
speed, air temperature, air pressure, precipitation, relative humidity, and vapor pressure
deficit were acquired from the daily, weekly, and monthly datasets on the FLUXNET





website. We analyzed the observations in the growing seasons from April to September for
the Northern Hemisphere and from October to March for the Southern Hemisphere. These
study periods were selected to avoid the high biases caused by the small solar radiation or the
extremely low evaporation ($\approx 0$) during the nongrowing season. The seasonal and annual data
were acquired by averaging the monthly data of the growing seasons. Following Ershadi et al.
(2014), the energy residual corrected latent heat fluxes were used, which means the residual
term in energy balance is attributed to the latent heat to force the energy balance closure. To
investigate the influence of different residual correction methods, the Bowen ratio energy
balance method was also adopted in the uncertainty analysis. In the Bowen ratio method, the
residual term is attributed into sensible heat and latent heat by preserving Bowen ratio (Twine
et al., 2000). The latent heat, sensible heat, and available energy ($R_n - G$) were restricted to
positive values (Han and Tian, 2018). The energy balance residual (W m$^{-2}$) and energy
balance closure ratio for each site are shown in Table S1.

The Nash‑Sutcliffe efficiency (NSE, Legates and McCabe, 1999) is used to evaluate the
efficiency of estimating $E$ by the two generalized complementary functions:

$$\text{NSE} = 1 - \frac{\sum(E - E_{est})^2}{\sum(E - \bar{E})^2} \tag{7}$$

where, $E_{est}$ (W m$^{-2}$) is the estimated evaporation according to equation (1) or equation (5) and
$\bar{E}$ is the mean value of $E$ (W m$^{-2}$).

**3. Results and discussion**
**3.1 Performance of the SGC function at multiple time scales**
The relationship between the estimated $E_{est}$ (site mean values) based on the SGC function
(equation (1)) and the observed $E$ of the 88 sites at multiple time scales is shown in Figure 1.
The regression equations and determination coefficients ($R^2$) were calculated by the site mean
results. Each dot in Figure 1 represents the site mean result averaged by daily (Figure 1a),
weekly (Figure 1b), monthly (Figure 1c), and yearly (Figure 1d) results, and the total
observation number is 88 (sites) at each timescale. Most of the results lie near the 1:1 line,
and all the regression slopes are close to 1 with high $R^2$ (0.95 ~ 0.99), which means the
sigmoid function exhibits good performance to estimate $E$ at multiple time scales. The
interceptions range from −1.69 to 2 W m$^{-2}$. All the coefficients of the regression show
indistinctive differences at different time scales. However, the evaluation merits show that the
performance varies at each time scale. The mean results of $NSE_H$, $R^2_H$, and $RMSE_H$ (the
subscript H corresponds to the sigmoid function proposed in Han and Tian, 2018) of these
sites are shown in Table 1. $R^2_H$ represents the mean value averaged by the determination
coefficients within each site. When the timescale changes from day to month, the mean $NSE_H$
increases from 0.33 to 0.55, and $R^2_H$ also increases from 0.61 to 0.75 (Table 1). However,
they both decrease at the annual scale ($NSE = 0.18$ and $R^2_H = 0.61$). These results indicate
that the SGC function exhibit the highest skill at the monthly scale. We inferred that there is a
tradeoff between the random error and the number of observations. $RMSE_H$ values decrease
from 24.56 W m$^{-2}$ at the daily scale to 7.33 W m$^{-2}$ at the annual scale, which means the
random error decrease as time scale increases. At the same time, the fewer observations at the
annual scale result in decreased variabilities of $x$ and $y$, which affect the performance of the
SGC function. On the other hand, Morton (1983) did not suggest using the complementary



principle for short time intervals (e.g., less than 3 days), mainly considering the lag times
associated with heat and water vapor change in the atmosphere, which can explain that the
weekly and monthly results are better than the daily results.

In previous studies, the SGC function was mainly applied at the daily scale. For example, the
results of Ma et al. (2015b) in the alpine steppe region showed that the NSE of the sigmoid
function is 0.26 at the daily scale, which is lower than our mean value in the grassland (0.73
$\pm$ 0.08). The RMSE (11.06 W m$^{-2}$) is smaller than ours (16.36 $\pm$ 1.48 W m$^{-2}$). The mean NSE
of 20 EC sites from the FLUXNET is 0.66 at daily scale in Han and Tian (2018), about two
times of the result in this study, and the RMSE (18.6 $\pm$ 0.94 W m$^{-2}$) is lower than our mean
result of 88 sites (24.56 $\pm$ 0.95 W m$^{-2}$).

The SGC function for the five selected sites of different ecosystem types is shown in Figure 2
to show the performance at multiple time scales (red lines in Figure 2). These five EC
monitoring sites were selected because they have long-period observations (> 10 years). The
five sites include an evergreen needle forest (CA-TP1, Figures 2(a) to (d)), a deciduous broad
forest (US-UMB, Figures 2(e) to (h)), a woody savanna (US-SRM. Figures 2(i) to (l)), a
cropland (US-Ne2, Figures 2(m) to (p)) and a grassland (US-Wkg, Figures 2(q) to (t)). As
observations decrease from daily to annual scale, the results converge on the middle part of
the sigmoid curves and lie closer to the fitted lines. For some sites, the annual results
concentrate on a narrow range with lower annual variabilities (e.g., Figures 2(h), 2(l) & 2(t)).
Generally, the key parameter ($b$) of the SGC function at these sites increases from the daily



scale to the annual scale, which indicates the sigmoid curves in the two-dimensional space of
$E_{rad}/E_{pen}$-$E/E_{pen}$ move upwards. The detailed discussion about the variation of the parameters
is elaborated in Section 3.4.

**3.2 Performance of the PGC function at multiple time scales**
The relationship between the estimated $E_{est}$ (site mean values) based on the PGC function
(equation (5)) and the observed $E$ of the 88 sites at multiple time scales is shown in Figure 3.
The slopes of the regression increase from 0.9 to 1 as the timescale changes from day to
month, and further increase to 1.01 at the annual scale. The intercept terms decrease from
13.06 W m$^{-2}$ at the daily scale to 0.01 W m$^{-2}$ at the monthly scale, and further decrease to
$-0.25$ W m$^{-2}$ at the annual scale. The R$^2$ values increase from 0.83 to 0.99 as time scale
increases. These coefficients of the regression show that the PGC function exhibit the highest
skill at the monthly scale. The mean values of NSE$_B$, R$^2$$_B$, and RMSE$_B$ (the subscript B
corresponds to the polynomial function proposed in Brutsaert, 2015) of these sites are shown
in Table 1. When the timescale changes from day to month, NSE$_B$ increases from 0.19 to
0.50, and R$^2$$_B$ increases from 0.61 to 0.75. They decrease at the annual scale (NSE = 0.25 and
R$^2$$_H$ = 0.63). Again, these evaluation merits indicate that the PGC function also exhibits the
highest skill at the monthly scale, which is the same as the SGC function.

The PGC function has been applied at multiple time scales in previous studies. Zhang et al.
(2017) evaluate the performance of the PGC function in estimating evaporation at 4 EC flux
sites located across Australia, and their results showed that the mean RMSE (24.67 W m$^{-2}$)


and $R^2$ (0.65) are close to our results (RMSE = 26.83 ± 1.16 W m$^{-2}$ and $R^2$ = 0.61) at the
daily scale. In Crago and Qualls (2018), the mean RMSE of 7 EC sites at the weekly scale is
20.6 W m$^{-2}$ and the mean $R^2$ is 0.81, which are close to our mean results (RMSE = 19.17 ±
0.95 W m$^{-2}$ and $R^2$ = 0.7).

The PGC functions for the five selected sites are also shown in Figure 2 (green lines). The
fitted lines almost duplicate with those of SGC function in most situations when $x$ is not too
high. However, they diverge from each other when $x$ becomes larger. Finally, $y$ exceeds 1
when $x$ is larger than $1/\alpha$. Generally, the key parameter ($c$) of the PGC function at these sites
decreases from daily scale to annual scale, which also indicates the fitted curves move
upwards.

**3.3 Performance comparison of the SGC and PGC functions**
The results of the 88 sites (Figure 1, Figure 3 and Table 1) show that the performance of the
two functions are similar at monthly and annual time scales, while the SGC function
performs slightly better than the PGC function at daily and weekly time scales. According to
the results in Figure 2, it can be recognized that the two functions with calibrated parameters
are approximately identical under non-humid environments, but their difference increases as
$x$ ($E_{rad}/E_{pen}$) increases. At daily and weekly time scales, quite a few ecosystems can produce
very high $E_{rad}/E_{pen}$. Specifically, 63 of the 88 sites have high $E_{rad}/E_{pen}$ ($x > 1/\alpha$) at the daily
scale and 24 sites have high values at weekly scale. However, there are only 3 sites with $x >$
$1/\alpha$ at the monthly scale and no site at the yearly scale. For the SGC function, in super humid





conditions, the upper part of the sigmoid curve is nearly flat and closer to the observations
(e.g., Figures 2 (a), (m) & (n)). However, for the PGC function, theoretically it cannot be
applied when $x$ is over $1/\alpha$ because the estimated $E_{est}$ will be higher than $E_{pen}$ which is
irrational. Thus, the sigmoid function performs slightly better at daily and weekly time scales.
But the difference vanished at the monthly scale as few high $E_{rad}/E_{pen}$ occurrences.

According to the results, the performance of the PGC function acts more sensitive to the
timestep than that of the SGC function. On one hand, the regression relationship between $E_{est}$
and the observed $E$ of the 88 sites shows the performance of the SGC function remains more
stable (Figure 1), while the regression results of the PGC function have higher variation when
the time scale changes (Figure 3). On the other hand, the estimation merits (Table 1) further
confirm the sensitivity of the PGC function. From daily scale to monthly scale, the increase of
$NSE_H$ is 0.22, while the increase of $NSE_B$ is 0.31; $RMSE_H$ decreases by 11.36 W m$^{-2}$ (46%)
and $RMSE_B$ decrease by 13.13 W m$^{-2}$ (49%). At the daily scale, quite a few ecosystems (63 of
88 sites) can experience frequent high $E_{rad}/E_{pen}$ ($> 1/\alpha$) occurrences, and the PGC function does
not have the ability to simulate $E$ accurately in this situation ($E_{est} > E_{pen}$) resulting in lower
efficiency. As time scale increases, the results converge on the middle part of the fitted line and
the number of high $x$ greatly reduces (Figure 2). Thus, the efficiency of the PGC function
increases obviously. It's the reason that the polynomial function acts more sensitive to the
timestep.

**3.4 Dependence of the key parameters of the SGC and PGC functions on time scales**


The key parameters of the two complementary functions ($b$ of the SGC function and $c$ of the
PGC function) vary at multiple time scales (Figure 2). To explore their changes, the values of
$1/b$ and $c$ of the 88 sites were averaged at each timescale. To take account of the situation that
$b$ is equal to infinity, we used $1/b$ instead of $b$ in this analysis. Figure 4 shows the change of
the two complementary functions with varied parameters at multiple time scales. The
averaged $1/b$ decreases from $0.45 \pm 0.05$ at the daily scale to $0.24 \pm 0.03$ at the annual scale
(Figure 4a), and the averaged $c$ decreases from $0.98 \pm 0.19$ at the daily scale (Figure 4b) to
$-0.37 \pm 0.22$ at the annual scale. The sign of $c$ changes from positive to negative at the
monthly scale.

We showed the histogram of $1/b$ and $c$ at multiple time scales in Figure 5 and Figure 6,
respectively. At the daily scale, half of the $1/b$ values are lower than 0.3 and the mean value
is $0.45 \pm 0.05$. At the weekly scale, the peak of the distribution moves left, nearly half of the
$1/b$ values are lower than 0.2 with the mean value of $0.36 \pm 0.04$. At the monthly scale, the
mean value is $0.29 \pm 0.04$ and the $1/b$ values continue to decrease. At the annual scale, the
mean value decreases to $0.24 \pm 0.03$ and 61% of the $1/b$ values are lower than 0.2. According
to Figure 6, at the daily scale, $c$ follows a normal distribution (p-value = 0.17, Kolmogorov-
Smirnov test) with the mean value of $0.98 \pm 0.21$. Nearly 1/3 of the $c$ values are lower than 0.
At the weekly scale, the center of the distribution moves left with the mean value of $0.43 \pm$
0.24. Half of the $c$ values are lower than 0. At the monthly scale, the mean value is $-0.04 \pm$
0.23, and 58% of the $c$ values are lower than 0. At the annual scale, the mean value decreases
to $-0.37 \pm 0.25$, and 63% of the $c$ values are lower than 0. These results support our





conclusion that $1/b$ and $c$ decrease as time scale increases. Generally, the distribution of $1/b$
and $c$ also move left within each ecosystem type according to Figures 5 and 6.

The reduction of $1/b$ and $c$ indicate the curves of the complementary functions move upwards
as time scale increases. Under non-humid conditions, the sigmoid function is a concave
function, which means:
$$\frac{1}{2}[f(x_1) + f(x_2)] > f(\frac{x_1 + x_2}{2}) \tag{8}$$

where, $f$ is the concave function, and $x_1$ and $x_2$ represent any two values on the x-axis. Since
most of the results follow the fitted line, the averaged results of longer timestep will go upwards
in the two-dimensional space of $E_{rad}/E_{pen}$-$E/E_{pen}$, so does the new fitted curve. Although under
the super humid condition, the SGC function is a convex function, there is fewer data in this
condition as time scale increases and the shape of this part is almost unchanged (Figure 4a). As
for the PGC function, when $x$ is in the range of 0 to $1/\alpha$, most part of it is a concave function.
For example, in the situation that $c$ is equal to 0, the second derivative is higher than 0 as long
as $x$ is lower than 2/3.

Furthermore, we found that the two key parameters, $b$ and $c$ present a significant correlation,
indicating the two functions can substitute each other in a sense. The relationship can be
described as: $1/b = 0.01c^2 + 0.11c + 0.24$ with $R^2$ higher than 0.96 at the monthly scale (Figure
5). The relationship keeps at other time scales with a slight difference in the regression
coefficients. At the daily scale, when $c$ is equal to 0, the corresponding $b$ is equal to 4.5, which
is the same as that of the theoretical derivation in Brutsaert (2015).






**3.5 Uncertainty analysis**

**3.5.1 Influence of ecosystem types**

The evaluation merits of the generalized complementary functions may differ among

ecosystem types. However, our results show that such variation generally not affect our

conclusion that the complementary functions perform best at the monthly scale. We show the

performance of the two functions at multiple timescales for each ecosystem type in Table S2.

Generally, the SGC function and the PGC function perform best at the monthly scale in most

ecosystem types (9 of 11) with the highest NSE and $R^2$, which is consistent with the overall

results. The exceptions include a closed shrubland site (CSH, N = 1) and evergreen broadleaf

forests (EBF, N = 8), in which the complementary functions perform not as well as in other

ecosystem types. The CSH site (IT-Noe) has the highest $NSE_H$ (0.11) and $NSE_B$ (0.12) at the

annual scale. In the EBF group, the highest $NSE_H$ (0.15) and $NSE_B$ (0.03) occur at the weekly

scale, but the $R^2$ values at the weekly scale ($R^2_H = 0.64$; $R^2_B = 0.62$) and those at the monthly

scale ($R^2_H = 0.62$; $R^2_B = 0.61$) are close. The RMSEs at the weekly scale are 14.95 W m$^{-2}$

and 16.08 W m$^{-2}$ for the sigmoid function and polynomial function, respectively, and those

values at monthly scale are 12.36 W m$^{-2}$ ($RMSE_H$) and 12.93 W m$^{-2}$ ($RMSE_B$). We inferred

the abnormal results of these two exceptions are related to the lower NSE values in these

ecosystem types. The mean NSE values at multiple time scale of CSH (−0.75) and EBF

(−0.66) are negative, while the values of the other ecosystem types are all positive.


**3.5.2 Performance at seasonal scale**





In consideration of the substantial discrepancy between the monthly results and the annual
results, we added an analysis at the seasonal scale, which is between the two timesteps. The
relationship between the estimated $E_{est}$ (site mean values) and the observed $E$ of the 88 sites
at seasonal scale is shown in Figure S1. For the SGC function, the regression result at the
seasonal scale is similar to that at the monthly scale (Figure S1a and Figure 1c). The values of
$NSE_H$ (0.33), $R^2_H$ (0.61), and $RMSE_H$ (10.16 W m$^{-2}$) at the seasonal scale are between the
monthly results and the yearly results (Table 1). For the PGC functions, the regression result
at the seasonal scale is extremely close to that at the yearly scale (Figure S1b and Figure 3d).
The evaluation merits ($NSE_B = 0.31$; $R^2_B = 0.63$; $RMSE_B = 9.94$ W m$^{-2}$) also range between
the monthly results and the yearly results (Table 1). These results indicate that the decline of
the model efficiency has already occurred at the seasonal scale and support our conclusion
that the complementary functions perform best at the monthly scale.

### 3.5.3 Influence of energy balance residual correction methods

So far, there are mainly two methods for surface energy closure correction in the
complementary studies. In the first method, the residual term is attributed into latent heat
directly as the "energy residual" (ER) closure correction (e.g., Ershadi et al., 2014; Han and
Tian 2018), which is adopted in above analysis. The second method is called the "Bowen
ratio" (BR) closure correction, in which the residual term is attributed into sensible heat and
latent heat by preserving Bowen ratio (e.g., Twine et al., 2000; Ma et al., 2015a). Based on
different correction methods, the evaluation results of the model performance may differ.
Thus, we recalculated our results by adopting the BR energy closure correction method. We





found the mean value of $1/b$ changes from $0.29 \pm 0.04$ (ER) to $0.40 \pm 0.05$ (BR) and the mean
value of $c$ changes from $-0.04 \pm 0.23$ (ER) to $0.63 \pm 0.24$ (BR) at monthly scale. It indicates
the key parameters could be affected by adopting different correction methods. However, the
results based on the BR method also support that the complementary functions perform best
on evaporation estimation at monthly scale (Table 2). The NSE and $R^2$ vales increase from
daily scale to monthly scale, and decrease from monthly scale to yearly scale, just following
the pattern showed in Table 1. Generally, the evaluation results based on the BR method are
worse than those based on the ER method. For example, when the ER method was replaced
by the BR method the NSE and $R^2$ values decrease ($\Delta NSE_H = -0.15$; $\Delta NSE_B = -0.23$; $\Delta R^2_H$
$= -0.07$; $\Delta R^2_B = -0.07$) and the RMSE values increase ($\Delta RMSE_H = 1.36$ W m$^{-2}$; $\Delta RMSE_B =$
$1.56$ W m$^{-2}$) at monthly scale. Ershadi et al. (2014) also found that the modeled $E_{est}$ values by
the PM equation, the AA approach and the modified Priestley-Taylor model (PT-JPL) show
higher agreement with the ER corrected evaporation instead of the BR corrected evaporation.
Ershadi et al. (2014) inferred the reason is that the observed sensible heat flux is more
reliable than the observed latent heat flux. The measurement of latent heat by the EC tower
may be confounded by minor instabilities when the boundary layer shrinks at night. To
summarize, although the different energy closure correction methods have some influences
on the key parameters and model efficiencies, they do not affect our conclusion that the
generalized complementary functions perform best at monthly scale.

**4. Conclusions**
In this study, evaporation estimation was assessed over 88 EC monitoring sites at multiple





time scales (daily, weekly, monthly, and yearly) by using two generalized complementary
functions (the SGC function and the PGC function). The performance of the complementary
functions at multiple time scales was compared, and the variation of the key parameters at
different time scales was explored. The main findings are summarized as follows:

(1) The sigmoid and polynomial generalized complementary functions exhibit the highest
skill in evaporation estimation at the monthly scale. The highest evaluation merits were
obtained at this time scale. The accuracy of the complementary functions highly depends on
the calculation timestep. The NSE increases from the daily scale (0.26, averaged by $NSE_H$
and $NSE_B$) to the weekly scale (0.37) and monthly scale (0.53) while decreases at the
seasonal scale (0.32) and the annual scale (0.22). The regression parameters between
estimated $E_{est}$ and observed site mean $E$ also support this conclusion for the PGC function.
The variations among different ecosystem types or between different energy balance
correction methods generally have no effect on this conclusion. Further evaporation
estimation studies by using the complementary functions can choose the monthly timestep to
achieve the most accurate results.

(2) The SGC function and the PGC function are approximately identical under non-humid
environments, while the SGC function performs better under super humid conditions implied
by high values of $x$ ($> 1/\alpha$) when the PGC function is theoretically useless ($E_{est} > E_{pen}$). At
daily and weekly time scales, quite a few ecosystems can experience frequent high $x$
occurrences and thus the SGC function performs slightly better than the PGC function at





these time scales. However, they perform very similarly at monthly and annual time scales as
few high $x$ occurrences. Besides, the performance of the PGC function is more sensitive to the
timestep than that of the SGC function.

(3) The key parameter $b$ of the SGC function increases and the key parameter $c$ of the PGC
function decreased as time scale increases. The value of $1/b$ is a quadratic function of $c$ with
higher $R^2$ ($> 0.96$). The relationship at the monthly scale can be described as: $1/b = 0.01c^2 +$
$0.11c + 0.24$. It indicates the two functions can substitute each other to some extent.

In this study, in order to find the most suitable time scale for applying the complementary
principle, the key parameters ($b$ and $c$) were calibrated to achieve the best model performance
at each timescale. Further studies on the prognostic application of the complementary
principle could focus on the reasonable prediction of the key parameters, and with the
predictable flexible parameters at different timescales, the complementary principle could be
integrated into hydrological models to reduce the uncertainty associated with evaporation
estimation.

**Code/Data availability**
All the data used in this study are from FLUXNET (http://fluxnet.fluxdata.org). The
intermediate data are available on request from the corresponding author
(tianfq@mail.tsinghua.edu.cn).





**Author contribution**


Songjun Han and Fuqiang Tian designed the experiments and Liming Wang carried them out.
Liming Wang developed the model code and performed the simulations. Liming Wang
prepared the manuscript with contributions from all co-authors.

**Competing interests**


The authors declare that they have no conflict of interest.

**Acknowledgements**


We are grateful for the financial support from National Science Foundation of China (NSFC
51825902, 51579249). We thank the scientists of FLUXNET (http://fluxnet.fluxdata.org) for
their generous sharing of their eddy flux data.



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



**List of Figure Captions**

**Figure 1.** The estimated evaporation based on the SGC function (equation (1)) vs the observed site mean evaporation at the daily scale (a), weekly scale (b), monthly scale (c) and yearly scale (d). Each dot represents the site mean result (N = 88 in each panel). The regression equations and determination coefficients ($R^2$) were calculated by the site mean results of the 88 EC sites.

**Figure 2.** Plots of $E/E_{pen}$ with respect to $E_{rad}/E_{pen}$ for five selected sites at multiple time scales. The black dots represent the observations; the red lines represent the SGC function; the green lines represent the PGC function; the blue lines are the P-T and Penman boundary lines. ENF, evergreen needleleaf forests; DBF, deciduous broadleaf forests; WSA, woody savannas; CRO, croplands; GRA, grasslands.

**Figure 3.** As in Figure 1 except for PGC function (equation (5)).

**Figure 4.** Plots of the SGC equation (1) with $\alpha = 1.26$ and varying $1/b$ values at multiple time scales (a). Plots of the PGC equation (5) with $\alpha = 1.26$ and varying $c$ values at multiple time scales (b). The blue lines are the P-T and Penman boundary lines.

**Figure 5.** Distribution of the key parameter $1/b$ at daily scale (a), weekly scale (b), monthly scale (c) and yearly scale (d): EBF, evergreen broadleaf forests (8); ENF, evergreen needleleaf forests (27); DBF, deciduous broadleaf forests (13); MF, mixed forests (5); Shrub (12), closed shrubland, open shrublands, woody savannas and savannas; CRO, croplands (6); WET, permanent wetlands (2).

**Figure 6.** Distribution of the key parameter $c$ at daily scale (a), weekly scale (b), monthly scale (c) and yearly scale (d): EBF, evergreen broadleaf forests (8); ENF, evergreen





needleleaf forests (27); DBF, deciduous broadleaf forests (13); MF, mixed forests (5); Shrub

(12), closed shrubland, open shrublands, woody savannas and savannas; CRO, croplands (6);

WET, permanent wetlands (2).

**Figure 7.** Relationships between $1/b$ and $c$ at the monthly scale.





**Table 1.** The evaluation merits (NSE, $R^2$ and RMSE in W m$^{-2}$) of the two generalized complementary functions using the "energy residual" (ER) closure correction method. The subscript H and B correspond to the SGC function proposed in Han and Tian (2018) and the PGC function proposed in Brutsaert (2015), respectively.

|  | Day | Week | Month | Season | Year |
|---|---|---|---|---|---|
| NSE$_H$ | 0.33 | 0.44 | 0.55 | 0.33 | 0.18 |
| NSE$_B$ | 0.19 | 0.3 | 0.50 | 0.31 | 0.25 |
| $R^2{}_H$ | 0.62 | 0.7 | 0.74 | 0.61 | 0.61 |
| $R^2{}_B$ | 0.61 | 0.7 | 0.75 | 0.63 | 0.63 |
| RMSE$_H$ | 24.56 | 17.67 | 13.20 | 10.16 | 7.33 |
| RMSE$_B$ | 26.83 | 19.17 | 13.70 | 9.94 | 6.96 |

**Table 2.** The evaluation merits (NSE, $R^2$ and RMSE in W m$^{-2}$) of the two generalized complementary functions using the "Bowen ratio" (BR) closure correction method. The subscript H and B correspond to the SGC function proposed in Han & Tian (2018) and the PGC function proposed in Brutsaert (2015), respectively.

|  | Day | Week | Month | Season | Year |
|---|---|---|---|---|---|
| NSE$_H$ | 0.01 | 0.23 | 0.4 | 0.17 | −0.07 |
| NSE$_B$ | −0.28 | 0.03 | 0.27 | 0.11 | −0.23 |
| $R^2{}_H$ | 0.53 | 0.62 | 0.67 | 0.54 | 0.52 |
| $R^2{}_B$ | 0.52 | 0.61 | 0.68 | 0.55 | 0.52 |
| RMSE$_H$ | 26.62 | 18.9 | 14.56 | 11.3 | 7.88 |
| RMSE$_B$ | 29.77 | 20.59 | 15.26 | 11.3 | 8.03 |

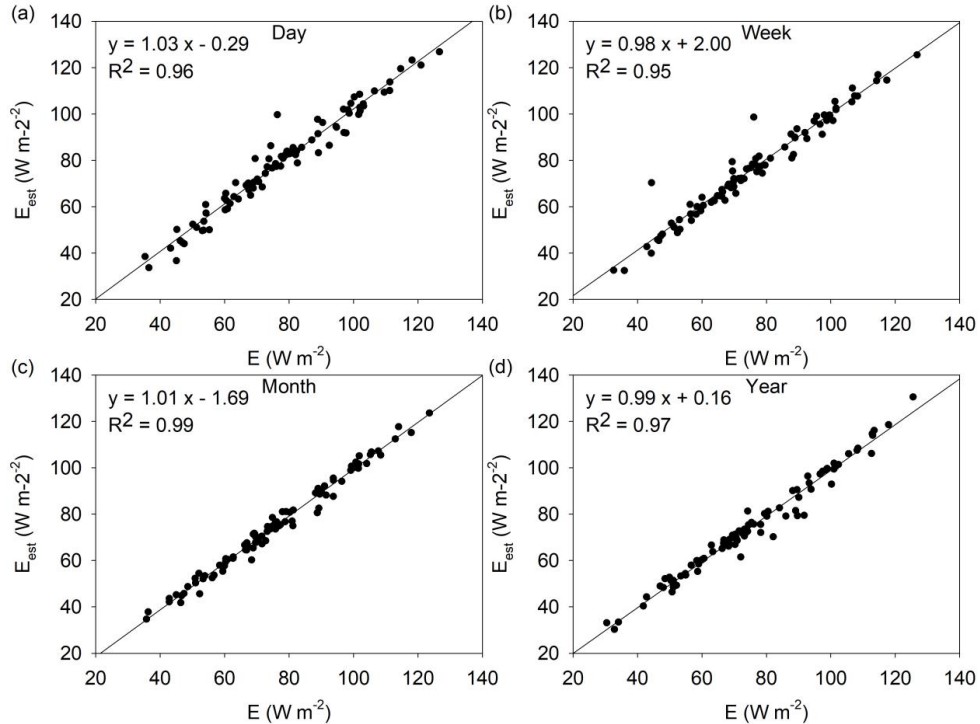

**Figure 1.** The estimated evaporation based on the SGC function (equation (1)) vs the

observed site mean evaporation at the daily scale (a), weekly scale (b), monthly scale (c) and

yearly scale (d). Each dot represents the site mean result (N = 88 in each panel). The

regression equations and determination coefficients ($R^2$) were calculated by the site mean

results of the 88 EC sites.

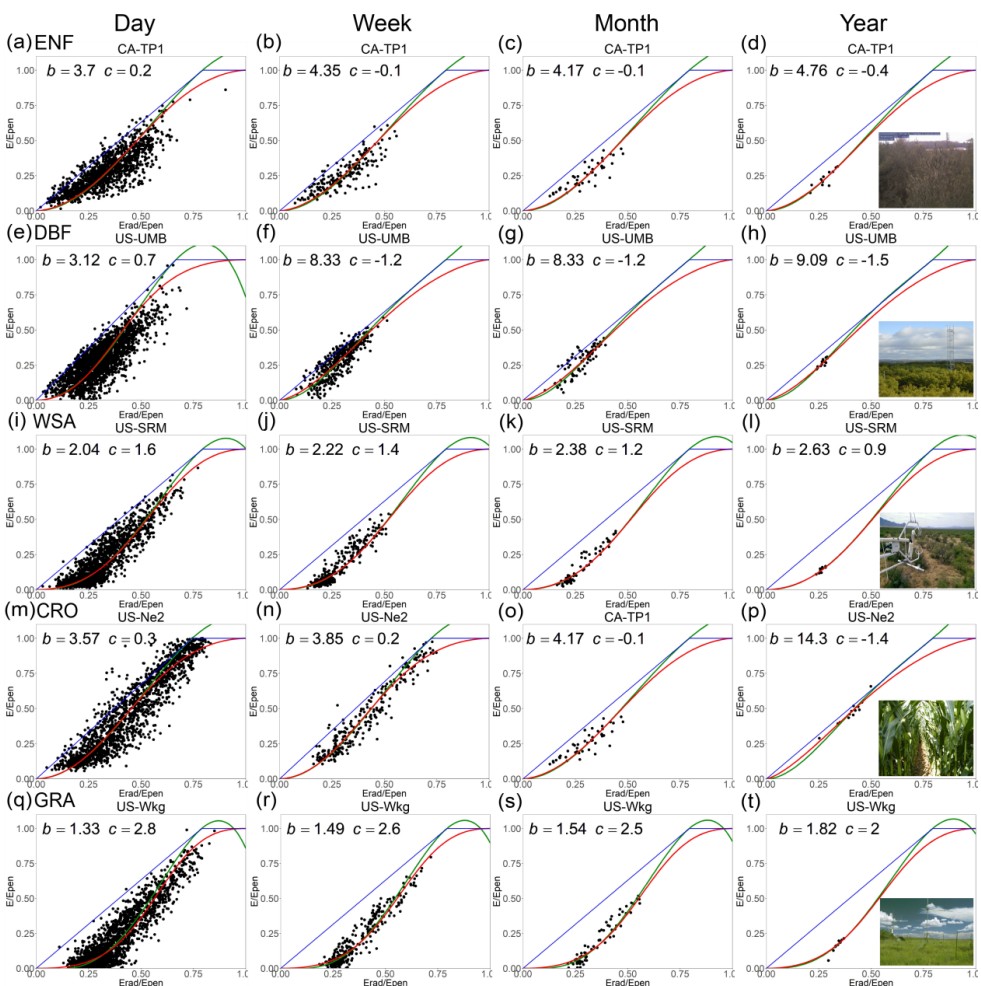

**Figure 2.** Plots of $E/E_{pen}$ with respect to $E_{rad}/E_{pen}$ for five selected sites at multiple time

scales. The black dots represent the observations; the red lines represent the SGC function;

the green lines represent the PGC function; the blue lines are the P-T and Penman boundary

lines. ENF, evergreen needleleaf forests; DBF, deciduous broadleaf forests; WSA, woody

savannas; CRO, croplands; GRA, grasslands.


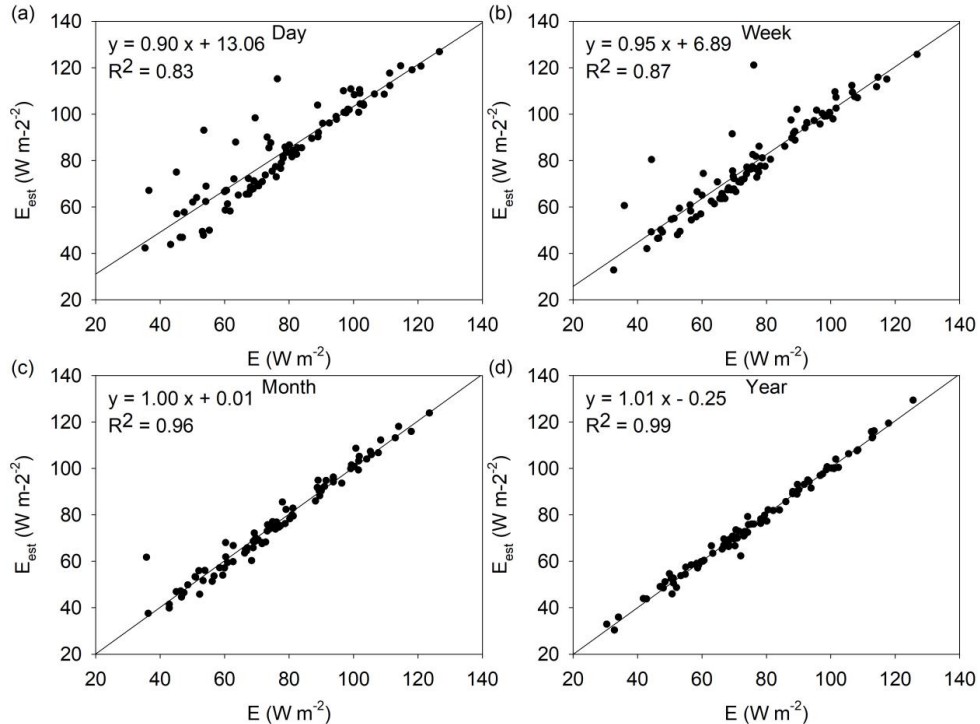

**Figure 3.** As in Figure 1 except for PGC function (equation (5)).





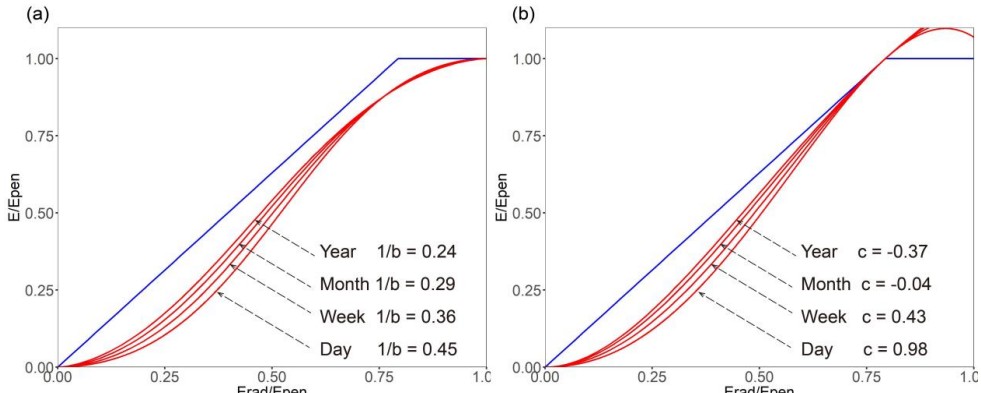

**Figure 4.** Plots of the SGC equation (1) with $\alpha = 1.26$ and varying $1/b$ values at multiple time

scales (a). Plots of the PGC equation (5) with $\alpha = 1.26$ and varying $c$ values at multiple time

scales (b). The blue lines are the P-T and Penman boundary lines.

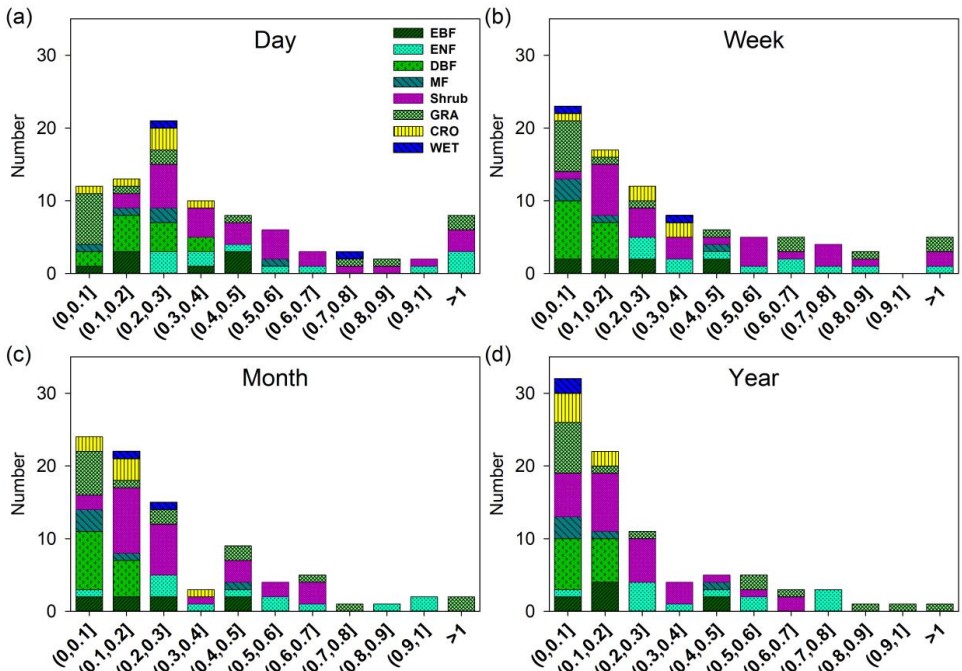

**Figure 5.** Distribution of the key parameter 1/$b$ at daily scale (a), weekly scale (b), monthly

scale (c) and yearly scale (d): EBF, evergreen broadleaf forests (8); ENF, evergreen

needleleaf forests (27); DBF, deciduous broadleaf forests (13); MF, mixed forests (5); Shrub

(12), closed shrubland, open shrublands, woody savannas and savannas; CRO, croplands (6);

WET, permanent wetlands (2).



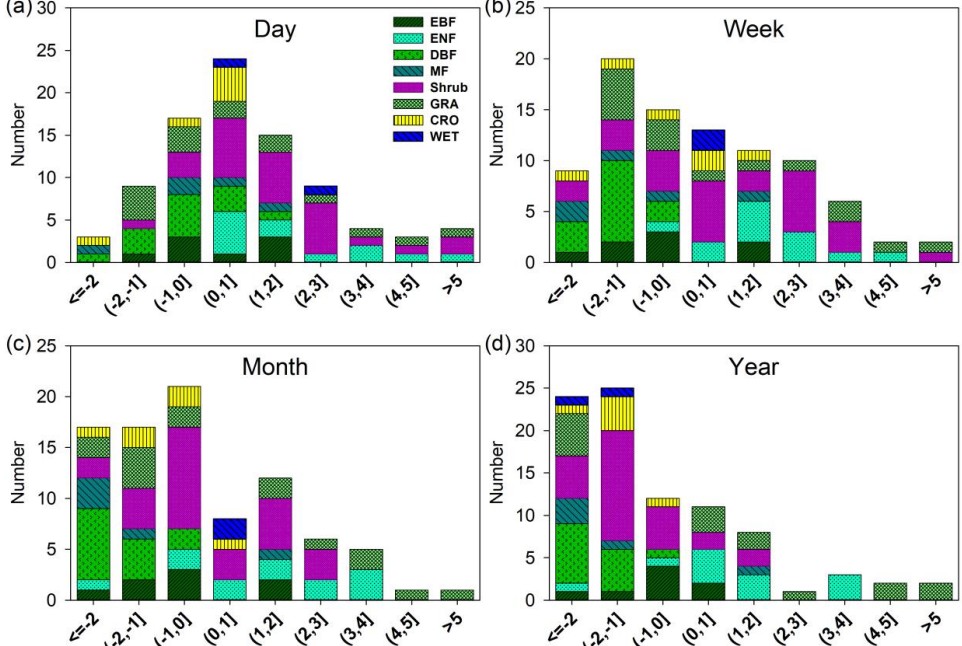

**Figure 6.** Distribution of the key parameter $c$ at daily scale (a), weekly scale (b), monthly

scale (c) and yearly scale (d): EBF, evergreen broadleaf forests (8); ENF, evergreen

needleleaf forests (27); DBF, deciduous broadleaf forests (13); MF, mixed forests (5); Shrub

(12), closed shrubland, open shrublands, woody savannas and savannas; CRO, croplands (6);

WET, permanent wetlands (2).





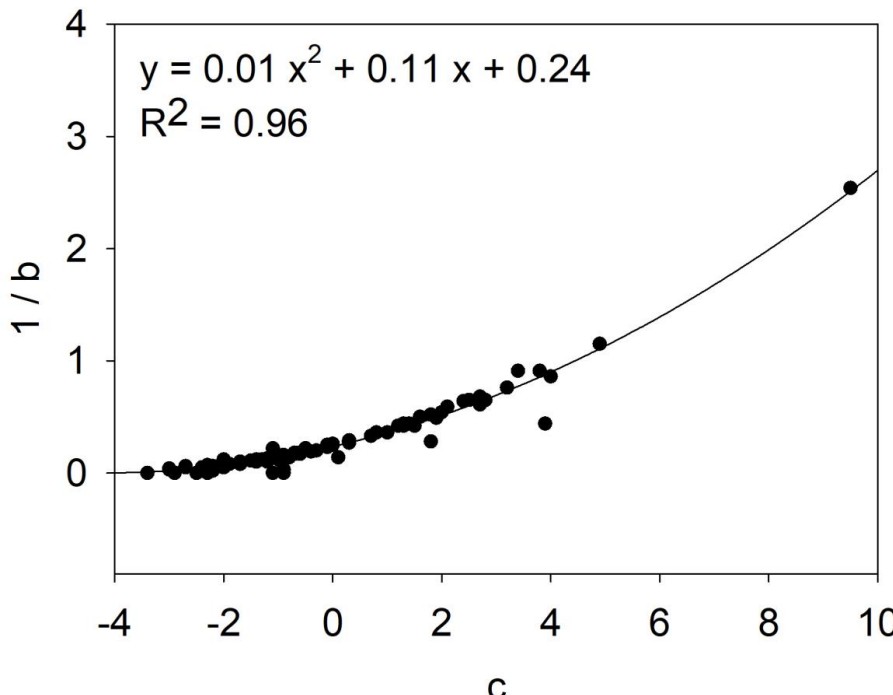

**Figure 7.** Relationships between $1/b$ and $c$ at the monthly scale.