# Peer review of "At which time scale does the complementary principle perform best on"

_Hydrology and Earth System Sciences, 2020_

## Referee Comment (RC1) · Anonymous Referee #1 · 18 Aug 2020

The MS is carelessly written. It should be thoroughly rechecked for grammar, typos, language constructs. For example, the AA method is mentioned several times before it is explained. Also, the first asymmetric AA method was of Kahler and Brutsaert (2006), and not by Brutsaert and Parlange (1998). Also, nobody reads the original work of Bouchet (1963), it seems, as it is in French. That may be the reason for frequent misquoting it. My understanding is that he never proposed a symmetrical CR. Even Brutsaert in his seminal book (1982) is controversial about this issue. The authors should clarify this issue though. I do not really see what we gain from this study. The high NSE value for the month comes about because its high variance between months and it is already being long enough to smooth things out. I bet that between Mays, Junes, Julys, etc., the NSE value would not be better than for the seasons and

years. The low value for the annual time-scale is a bit worrisome as it means that these two chosen methods cannot replicate any long-term trends in ET rates to acceptable accuray, which diminishes their potental values for long-term hydrological modeling.

---

## Author Comment (AC1) · 23 Aug 2020

Response to interactive comment on "At which time scale does the complementary principle perform best on evaporation estimation?" by Liming Wang et al.

Anonymous Referee #1

General response: Thank you for the timely review. We are very happy to hear the critical voice although we do not agree with many of them. We would like to discuss these contradictions with the reviewers. In the following we provided point-by-point responses as follows.

-The MS is carelessly written. It should be thoroughly rechecked for grammar, typos, language constructs.

[Figure]

Response: Thank you for the comments. We will go through and revise the manuscript thoroughly and hire some language experts to help polish the manuscript again.

-For example, the AA method is mentioned several times before it is explained.

Response: Thank you for pointing out this problem. we had provided the full name for it when the first time it is mentioned (Line 54-56, hereafter all lines numbers are based on the tracked version). Also, we moved the explanation from the methodology part to the introduction part.

-Also, the first asymmetric AA method was of Kahler and Brutsaert (2006), and not by Brutsaert and Parlange (1998).

Response: According to our reading, Brutsaert and Parlange (1998) provided the following equation in their paper: E=[(1+b) E0 - a Epa ]/b where, E0 has the same meaning of Epo in our manuscript (i.e., potential evaporation), and a is a pan coefficient, b is an asymmetric parameter. Our statement "the CR was extended to a linear function with an asymmetric parameter (Brutsaert and Parlange, 1998)" refers to this equation.

Kahler and Brutsaert (2006) summarized the previous work of Brutsaert and Stricker (1979), Brutsaert and Parlange (1998), and Brutsaert (2005) and gave the equation:

(1+b) E0=Cp Epa+bE where, Cp is a constant parameter. We can see that this equation holds the same format with Brutsaert and Parlange (1998) after appropriate transformation (and replacing Cp with a). It may be the first time it was called "asymmetric AA". Thank you.

-Also, nobody reads the original work of Bouchet (1963), it seems, as it is in French. That may be the reason for frequent misquoting it. My understanding is that he never proposed a symmetrical CR. Even Brutsaert in his seminal book (1982) is controversial about this issue. The authors should clarify this issue though.

Response: Yes, the original work of Bouchet (1963) is French. In our institute of Tsinghua University, we have a PhD student coming from France, and he had translated this paper into English several years ago. We are pleased to provide the English version of Bouchet (1963) at the end of the response (Supplement) for the reference. After reading this paper, we suggest that the contribution of Bouchet (1963) should be respected.

Equation (5) and Figure 2 of Bouchet (1963) show a symmetrical complementary relationship: ETP+ETR=2 ETP0 where, ETR is the energy corresponding to the real evapotranspiration, ETP is corresponding to Epa, and ETP0 is corresponding to Epo.

In the book of Brutsaert (1982, p224-225), the above equation is cited as equation (10.35), and Brutsaert said that Bouchet (1963) arrived at the complementary relationship and admit Bouchet's approach contains worthwhile ideas and led to further developments. Brutsaert thought this method is not used widely because the assumption is strict and it did not provide exactly measures of Epa and Epo.

Thank you.

-I do not really see what we gain from this study. The high NSE value for the month comes about because its high variance between months and it is already being long enough to smooth things out.

Response: The aim of this study is to investigate at which time scale the complementary principle performs best on evaporation estimation. Based on this reviewer's comment, we understand that the reviewer gained that complementary functions perform best at the monthly scale. Actually, it's exactly what we want to convey to the audience. We did not find the evidence in previous studies or theoretical derivation which had already revealed this conclusion. Without these results, it is still uncertain how long is "enough to smooth things out". It could be 7 days, 30 days or 90 days. We agree with the reasons for the high NSE value at the monthly scale given by the reviewer, these reasons are also discussed in our manuscript (Line 236- 241). The "high variance" can be corresponding to our explanation about "variabilities of x and y" (Line 240), and the "smooth things out" can be corresponding to our explanation of RMSE. Thank you.

-I bet that between Mays, Junes, Julys, etc., the NSE value would not be better than for the seasons and years.

Response: We are not very clear about this comment. In the current version, the study periods are from April to September for the Northern Hemisphere and from October to March for the Southern Hemisphere. Did the reviewer mean that if the study periods are shortened (e.g, from May to July), the NSE values at the monthly scale will be worse than for the seasons and years? We have provided the results for May to July in Table R1. In this situation, the seasonal result is equal to the annual result and there is one seasonal result (May to July) each year. These results still support our conclusion. The NSE values at the monthly scale (NSEH = 0.38 and NSEB = 0.32) are higher than those at the seasonal/annual scale (NSEB = −0.07 and NSEB = −0.05). Thank you for providing an opportunity to test the uncertainty in the length of study periods.

Table R1. The evaluation merits (NSE, R2 and, RMSE in W m−2) of the two generalized complementary functions from May to July (The table is better viewed in the supplement) ãĂĂ Month Season/Year NSEH [0.38] [−0.07] NSEB [0.32] [−0.05] R2H [0.63] [0.56] R2B [0.63] [0.56] RMSEH [12.17] [8.86] RMSEB [21.51] [8.81]

-The low value for the annual time-scale is a bit worrisome as it means that these two chosen methods cannot replicate any long-term trends in ET rates to acceptable accuracy, which diminishes their potential values for long-term hydrological modeling.

Response: Yes, the complementary functions perform worse in estimating E at the annual scale. To the best of our knowledge, this point had not been thoroughly discussed previously. We did not recommend choosing the annual scale as the timestep to estimate E because of the low efficiency. However, we can still replicate the long-term trends in E rates by adopting the monthly timestep. Thank you.

The response file can be found in Supplement. But the revised manuscript was not submitted as supplement following the introduction of the journal.
References Bouchet, R. J.: Evapotranspiration réelle et potentielle, signification clima-tique. IAHS Publ, 62, 134-142, 1963. Brutsaert, W.: Evaporation into the atmosphere: theory, history and applications. Springer, New York, 1982 Brutsaert, W.,: Hydrology: An Introduction, 605 pp., Cambridge Univ. Press, New York. 2005 Brutsaert, W., Par-lange, M. B.: Hydrologic cycle explains the evaporation paradox. Nature, 396(6706), 30-30, 1998. https://doi.org/ 10.1038/23845 Brutsaert, W., Stricker, H.: Advection-Aridity approach to estimate actual regional evapotranspiration. Water Resour. Res., 15(2), 443-450, 1979. https://doi.org/ 10.1029/WR015i002p00443 Hobbins, M.T., Ramirez, J.A., Brown, T.C.: Trends in pan evaporation and actual evapotranspiration across the conterminous US: Paradoxical or complementary? Geophys. Res. Lett., 31(13), 2004. https://doi.org/10.1029/2004GL019846 Kahler, D. M., Brutsaert, W.: Complementary relationship between daily evaporation in the environment and pan evaporation. Water Resour. Res, 42(5), 2006. https://doi.org/10.1029/2005WR004541 Penman, H. L.: Natural evaporation from open water, bare soil and grass. Proc. R. Soc. London, Ser. A., 193(1032), 120-145, 1948. https://doi.org/10.1098/rspa.1948.0037 Priestley, C. H. B., Taylor, R. J.: On the assessment of surface heat-flux and evap-oration using large-scale parameters. Mon. Weather Rev., 100(2), 81-92, 1972. https://doi.org/10.1175/1520-0493(1972)100<0081:Otaosh>2.3.Co;2

Please also note the supplement to this comment:
https://hess.copernicus.org/preprints/hess-2020-379/hess-2020-379-AC1-supplement.pdf
* * *
[Figure]

**Supplement:**

(Reviewers comments in Italic and responses in upright Roman)

**Anonymous Referee #1**

General response: Thank you for the timely review. We are very happy to hear the critical voice although we do not agree with many of them. We would like to discuss these contradictions with the reviewers. In the following we provided point-by-point responses as follows.

*-The MS is carelessly written. It should be thoroughly rechecked for grammar, typos, language constructs.*

Response: Thank you for the comments. We will go through and revise the manuscript thoroughly and hire some language experts to help polish the manuscript again.

*-For example, the AA method is mentioned several times before it is explained.*

Response: Thank you for pointing out this problem. we had provided the full name for it when the first time it is mentioned (**Line 54-56, hereafter all lines numbers are based on the tracked version**). Also, we moved the explanation from the methodology part to the introduction part.

*-Also, the first asymmetric AA method was of Kahler and Brutsaert (2006), and not by Brutsaert and Parlange (1998).*

Response: According to our reading, Brutsaert and Parlange (1998) provided the following equation in their paper:

$$E = \left[(1 + b)E_0 - aE_{pa}\right]/b$$

where, $E_0$ has the same meaning of $E_{po}$ in our manuscript (i.e., potential evaporation), and $a$ is a pan coefficient, $b$ is an asymmetric parameter. Our statement "the CR was extended to a linear function with an asymmetric parameter (Brutsaert and Parlange, 1998)" refers to this equation.

Kahler and Brutsaert (2006) summarized the previous work of Brutsaert and Stricker (1979), Brutsaert and Parlange (1998), and Brutsaert (2005) and gave the equation:

$$(1 + b)E_0 = C_p E_{pa} + bE$$

where, $C_p$ is a constant parameter. We can see that this equation holds the same format with Brutsaert and Parlange (1998) after appropriate transformation (and replacing $C_p$ with $a$). It may be the first time it was called "asymmetric AA". Thank you.

*-Also, nobody reads the original work of Bouchet (1963), it seems, as it is in French. That may be the reason for frequent misquoting it. My understanding is that he never proposed a symmetrical CR. Even Brutsaert in his seminal book (1982) is controversial about this issue. The authors should clarify this issue though.*

Response: Yes, the original work of Bouchet (1963) is French. In our institute of Tsinghua University, we have a PhD student coming from France, and he had translated this paper into English several years ago. We are pleased to provide the English version of Bouchet (1963) at the end of the response for the reference. After reading this paper, we suggest that the contribution of Bouchet (1963) should be respected.

Equation (5) and Figure 2 of Bouchet (1963) show a symmetrical complementary relationship:

$$ETP + ETR = 2ETP_0$$

where, ETR is the energy corresponding to the real evapotranspiration, ETP is corresponding to $E_{pa}$, and $ETP_0$ is corresponding to $E_{po}$.

In the book of Brutsaert (1982, p224-225), the above equation is cited as equation (10.35), and Brutsaert said that Bouchet (1963) arrived at the complementary relationship and admit Bouchet's approach contains worthwhile ideas and led to further developments. Brutsaert thought this method is not used widely because the assumption is strict and it did not provide exactly measures of $E_{pa}$ and $E_{po}$.

Thank you.

*-I do not really see what we gain from this study. The high NSE value for the month comes about because its high variance between months and it is already being long enough to smooth things out.*

Response: The aim of this study is to investigate at which time scale the complementary principle performs best on evaporation estimation. Based on this reviewer's comment, we understand that the reviewer gained that complementary functions perform best at the monthly scale. Actually, it's exactly what we want to convey to the audience. We did not find the evidence in previous studies or theoretical derivation which had already revealed this conclusion. Without these results, it is still uncertain how long is "enough to smooth things out". It could be 7 days, 30 days or 90 days. We agree with the reasons for the high NSE value at the monthly scale given by the reviewer, these reasons are also discussed in our manuscript (**Line 236- 241**). The "high variance" can be corresponding to our explanation about "variabilities of *x* and *y*" (**Line 240**), and the "smooth things out" can be corresponding to our explanation of RMSE. Thank you.

*-I bet that between Mays, Junes, Julys, etc., the NSE value would not be better than for the seasons and years.*

Response: We are not very clear about this comment. In the current version, the study periods are from April to September for the Northern Hemisphere and from October to March for the

Southern Hemisphere. Did the reviewer mean that if the study periods are shortened (e.g, from May to July), the NSE values at the monthly scale will be worse than for the seasons and years? We have provided the results for May to July in Table R1. In this situation, the seasonal result is equal to the annual result and there is one seasonal result (May to July) each year. These results still support our conclusion. The NSE values at the monthly scale ($NSE_H$

= 0.38 and $NSE_B$ = 0.32) are higher than those at the seasonal/annual scale ($NSE_B$ = −0.07

and $NSE_B$ = −0.05). Thank you for providing an opportunity to test the uncertainty in the length of study periods.

**Table R1.** The evaluation merits (NSE, $R^2$ and, RMSE in W m$^{-2}$) of the two generalized complementary functions from May to July

|  | Month | Season/Year |
| --- | --- | --- |
| $NSE_H$ | 0.38 | −0.07 |
| $NSE_B$ | 0.32 | −0.05 |
| $R^2_H$ | 0.63 | 0.56 |
| $R^2_B$ | 0.63 | 0.56 |
| $RMSE_H$ | 12.17 | 8.86 |
| $RMSE_B$ | 21.51 | 8.81 |

*-The low value for the annual time-scale is a bit worrisome as it means that these two chosen*

*methods cannot replicate any long-term trends in ET rates to acceptable accuracy, which*

*diminishes their potential values for long-term hydrological modeling.*

Response: Yes, the complementary functions perform worse in estimating $E$ at the annual scale. To the best of our knowledge, this point had not been thoroughly discussed previously.

We did not recommend choosing the annual scale as the timestep to estimate $E$ because of the low efficiency. However, we can still replicate the long-term trends in $E$ rates by adopting the monthly timestep. Thank you.

**References**

Bouchet, R. J.: Evapotranspiration réelle et potentielle, signification climatique. IAHS Publ, 62, 134-142,

1963.

Brutsaert, W.: Evaporation into the atmosphere: theory, history and applications. Springer, New York, 1982

Brutsaert, W.,: Hydrology: An Introduction, 605 pp., Cambridge Univ. Press, New York. 2005

Brutsaert, W., Parlange, M. B.: Hydrologic cycle explains the evaporation paradox. Nature, 396(6706),

30-30, 1998. https://doi.org/ 10.1038/23845

Brutsaert, W., Stricker, H.: Advection-Aridity approach to estimate actual regional evapotranspiration.
Water Resour. Res., 15(2), 443-450, 1979. https://doi.org/ 10.1029/WR015i002p00443

Hobbins, M.T., Ramirez, J.A., Brown, T.C.: Trends in pan evaporation and actual evapotranspiration across
the conterminous US: Paradoxical or complementary? Geophys. Res. Lett., 31(13), 2004.
https://doi.org/10.1029/2004GL019846

Kahler, D. M., Brutsaert, W.: Complementary relationship between daily evaporation in the environment
and pan evaporation. Water Resour. Res, 42(5), 2006. https://doi.org/10.1029/2005WR004541

Penman, H. L.: Natural evaporation from open water, bare soil and grass. Proc. R. Soc. London, Ser. A.,
193(1032), 120-145, 1948. https://doi.org/10.1098/rspa.1948.0037

Priestley, C. H. B., Taylor, R. J.: On the assessment of surface heat-flux and evaporation using large-scale
parameters. Mon. Weather Rev., 100(2), 81-92, 1972.
https://doi.org/10.1175/1520-0493(1972)100<0081:Otaosh>2.3.Co;2

**REAL EVAPOTRANSPIRATION AND POTENTIAL CLIMATIC SIGNIFICANCE**

**R . J . BOUCHET**
Central station of bioclimatologie, Versailles

National institute of the Agronomic research (France)

The real evapotranspiration of an area represents the water really lost in the form of vapor, the potential evapotranspiration, the water likely to be lost under the same conditions when it is not limiting factor any more. The knowledge of these two data is obviously indispensable to study the circulation of water or to define the needs for the water of the cultures.

We propose to show the connections that exist not only between ETP and ETR, but also between these terms and the various elements of the energy report (the total radiation, the radiation of long wave, etc…), by using the method of the energy assessment. The simple relations that we will establish will permit to better define the climatic significance of ETR and ETP. It will then be possible to specify their respective variations when we will try to modify the climate in more or less vast zones, either by irrigating, or by changing the cover of the ground.

**I -- STARTING ASSUMPTION -- SCALE OF THE ASSESSMENT**

The study of the energy assessment supposes the preliminary definition of the system limits. To avoid taking into account the phenomena of accumulation and restitution of heating during the diurnal and night phases, the assessment will relate to one 24-hour period, the variations of temperature are then generally negligible.

The system includes the whole of the vegetable mass, a superficial section of ground, and a lower section of the atmosphere. Dimensions of these sections are just as the nycthemeral variations of temperature remain appreciable. The system exchange of heating with outside during this period takes place without the phenomena of radiation and evaporation, by conducting in deep layers of the ground (Qs) and by convecting (Qa) towards the high layers of atmosphere.

If this system itself is located in a zone that does not present the same climatic characteristics for various reasons, there will be the side exchanges of energy on the walls which has to be analyzed.

The side exchanges by conduction in the ground are negligible. It is not the same side exchange as in the atmosphere due to the movements of the standardized mass of air which we will indicate under the general name "of the oasis effects". Given the heterogeneity of a point to another of the type ground, the vegetable cover, the phenomena of evaporation, side movements of energy or "the oasis effect" are the rules under the natural conditions.

We can schematically represent the phenomenon of the oasis effect of the following manner (Fig. 1). If in a flat and homogeneous zone, an heterogeneity appears (the characteristics of the ground such as the thermal conductibility, the specific heating, the moisture or the nature of vegetable cover, the different ETR, etc…), it develops in the direction of the air circulation a disturbed zone where the medium factors find to be modified compared to the general climate because of heterogeneity. The oasis effect thus corresponds to an intrusion of the external system on the studied system, not only by its immediate edges but by the whole of the limit of the disturbed zone.

[Figure]

Fig. 1 -- Importance of the disturbed climatic zone according to the dimension of the heterogeneous system compared to the external system.
hauteur --- height;
climat général du système extérieur    --- the general climate of external system
limite de l'effet d'oasis --- the limit of the effect of oasis
zone climatique perturbeé --- the perturbed climatic zone
surface du sol --- the ground surface
système extérieur --- the external system

The disturbance rises all the more in height since the heterogeneous zone is extended. It is always presented in the form of a "flat lens" for which the thickness is weak compared to horizontal dimensions. We can thus define, for each meteorological scale and each scale of heterogeneity, an oasis effect of corresponding scale (table 1) which gives the side exchange of energy $Q_1$ $Q_2$ $Q_3$ $Q_4$ $Q_5$. These are the exchanges that we will try to specify later on in the equation of the energy assessment and which we must take into account to define horizontal dimensions to give to the system.

As we propose to connect ETR to the different terms of the energy assessment, we must consider ETR as uniform on all the surfaces of the system. Thus it comes to determine from which minimum surface, the real evapotranspiration can affect the climatic factors that we use to define the climate, by acting on the energy assessment. It is only when we attain this minimum surface that we will be ensured to have an excellent connection between the climatic factors ($\theta_a$, $\theta_r$, wind, etc.) and ETR, since these factors will not only be considered any more as a more or less direct possible cause, but also as an effect. The minimum zone presenting the character of uniformity will have to thus be just as the disturbance reaches the level to which one refers to have the climatic data. Those are collected to 2 m above the ground with instruments having time-constants of the order of a minute. We will thus, a priori, have to consider only the thermal phenomena having a higher scale or equal to that of turbulence itself ($> e_2$).

The heating exchange of greater scale (e 3 4 5) are integrated in Qa term of the energy assessment. They thus contribute to define the climate. Thus, the "oasis effects" of great scale such as those existing between the maritime zones and the continents are found in the climatic data of the meteorological networks. In the same way, on the scale of the 1/2 day, the breezes of sea or ground can be treated as the oasis effects of higher scale or equal to e3.

The heating exchange related to the scales lower than e2 are from the concepts even of the negligible scale and can be regarded as the simple movements of standardization within the system which does not affect the climate just as we define it.

To respect the scale of turbulence, the zone considered for the energy assessment should thus take the character of uniformity on the distances of a few hundreds of meters, to see a few kilometers. The minimum extent on the surface is thus of the order from 10 to 100 ha.

| TABLE 1 | | | |
|---|---|---|---|
| Scale (symbol in the text) | Scale of time | Scale of distance | Corresponding oasis effect (symbol in the text) |
| Molecular      e1 | 10(-9) second | | Q1 |
| Turbulent      e2 | 1s. to several minutes | A few hundreds of meters | Q2 |
| Associated convection and movements      e3 | 10 minutes to several hours | Several kilometers | Q3 |
| | 3 to 4 days | 1000 to 2000 km | Q4 |
| Cyclonic      e4 | | | |
| | 10 to 30 days | 5000 to 10,000 km | Q5 |
| Planetary          e5 | | | |

We define in Meteorology the scales of turbulence which permit to neglect the phenomena whose scale is small compared to the macroscopic movement considered.

On the whole of this surface, ETR will be uniform by hypothesis. If the real evapotranspiration around this system is different from that located inside, there will be the
oasis effects; those will be all the more important scale because the zone considered will be
large; within the framework of our definition, they will be lower or equal to turbulence. The
potential evapotranspiration will be thus variable within the system according to the distance
to these edges. The potential evapotranspiration then will be considered in the center of the
device, where it is weakest or strongest. If the surface of the system were more important,
ETP in the center would decrease or increase, but the climatic factors as they are generally
considered, would then start to be affected, which would be against the starting hypothesis
since we propose to define VETP of the initial climate and not Y ETP modified by the
variations of evaporation.

In conclusion, ETP can be defined in the level of the meteorological shelter to 2 m
only like the potential evapotranspiration in the center of a uniform zone at the view point of
ground, vegetation or evaporation and at least few tens of hectares.

Thus, the reasoning which will follow will not be able to apply strictly to even the
homogeneous zones which do not have the sufficient size and a fortiori, to the heterogeneous
zones. We will thus encounter the great difficulties in defining ETR or ETP as the climatic
factors in the zones of transition, because the oasis effects of scale lower than that used to
define the climate, will modify the suggested equalities. These cases will meet in particular
for the complex checkerwork that represents the vegetation of a zone of mixed-farming, for
the small oases in arid zone, the clearings in the forest zones, for the edges of the massive
forest or the maritime coasts. However, the suggested equations provide an approach for the
problem.

II – ASSESSMENT Of ENERGY (*)

Suppose an uniformed system corresponding to the preceding conditions. During a period
of 24 hours, the energy assessment brought to the unit of area is,

(1) $$(1 - a)\, Rg + (1 - a')\, Ra - a''\, \sigma T^4 + Q = E - C = ETR$$

the equation which gives the real evapotranspiration of the area considered. Thus, ETR more
or less limited by the intervention of the factor "water" plays an essential part in the
interaction of the physical data of the climate by these terms $\sigma T^4$, Ra and in certain
measurement Ra or even Rg.

ETP corresponds, by definition, in case the available energy is the only factor limiting
the evaporation. Study the passage from ETR to ETP in the previously defined system. Let us
indicate by ETP0 the value of the potential evapotranspiration when ETR is equal to ETP.
Suppose this condition to be realized

(2) $$ETR = ETP = ETP_0$$

Admit that for an independent reason of the energy phenomena, ETR decreases. This case could result in a period of dryness, of the maturity of vegetation, of its cut, etc…The reduction in ETR releases an energy q1 such as,

$$(3) \qquad\qquad ETP_0 - ETR = q_1.$$

With the scale considered, this modification of balance inside of the system does not affect the total radiation and only intervenes very slightly on the Ra term via the temperature and the moisture of the low atmospheric layers. The only important modification which will bring to the temperature and the turbulence, will cause a modification of ETP. Under the best conditions, i.e. if the transformation does not modify the exchanges of the system with outside, the energy returning available (q1) should correspond to an increase of ETP. Thus, without the modification of the initial climate from the energy point of view and in particular without the variation of the different primitive oasis effects, we will have

$$(4) \qquad\qquad ETP = ETP_0 + q_1$$

Where, by considering (3),

$$(5) \qquad\qquad ETP + ETR = 2\,ETP_0$$

(*) We will admit as positive the energy received on the surface of the ground, as negative the energy lost. The following symbols will be used:

a      ---   albedo, the reflection fraction of the total radiation (expressed in percentage)
a'     ---   the reflection fraction of the atmospheric radiation
a''    --- the emissivity
Rg    --- the total radiation (the solar radiation $\leq 5\mu$   received on an horizontal surface)
Ra    ---   the atmospheric radiation of the long wave $> 5\mu$
$\sigma T^4$ – the radiation of the ground at the absolute temperature T with an emissivity equal to the unit
E      ---   the energy involved by the evaporation
C      ---   the energy involved by the condensation
Q      ---   the energy exchanged by the conduction-convection by the considered system with outside
Qs    ---   the energy exchanged by the conduction in the ground
Qa    ---   the energy exchanged by the conduction-convection in the air. Qa comprises the exchange of heating of the various scales
ETR --- the energy corresponding to the real evapotranspiration
ETP ---    the energy corresponding to the potential    evapotranspiration

Thus, for a given climate, all would occur as if there were symmetry between ETR and ETP compared to a constant ETP0. Very generally, the transformation will not occur without the modification of the exchanges with outside and the equalities will transform themselves into inequalities.

$$(6) \qquad\qquad ETP + ETR \leqslant 2\,ETP_0$$

By using the equality (5) and by clarifying the values of Q according to the scales, the general equation (1) can be written as,

$$(7) \qquad ETP + (1-a)\,Rg + (1-a')\,Ra - a''\sigma T^4 + Qs + Q_3 + Q_{4.5} = 2\,ETP_0$$

(1 - a) Rg', ETP0, $Q_{4.5}$ are not affected by the relative variation of ETR, and ETP related to the availability of the water. $\sigma T^4$, Ra, Qs and even $Q_3$ are on the contrary variable. We can thus put (7) in the following form, by grouping the variable terms in a function g,

$$(8) \qquad\qquad ETP = 2\,ETP_0 + g - (1-a)\,Rg - Q_{4.5}$$

and according to (5),

$$(9) \qquad\qquad ETR = (1-a)\,Rg' + Q_{4.5} - g.$$

For the given values ETP0, Rg', $Q_{4.5}$ , ETR is a decreasing function of g, then ETP is an increasing function. When ETR = 0, ETP takes the maximum value corresponding to 2ETP0. Moreover, according to (9),

$$(10) \qquad\qquad g = (1-a)\,Rg' + Q_{4.5}$$

When the water is not a limiting factor, ETR becomes by definition equal to ETP. The variation of ETR explains then that of ETP. The maximum value likely to be taken under these conditions by ETR corresponds to the possible maximum value of ETP. According to (9), ETR will be maximal when g will be null. In fact, g that essentially represent the net radiation of the long wave ($\sigma T^4$ - Ra), engine of night cooling, could not be positive, otherwise the night amplitude of the temperature would change the sign and the night temperatures would be increasing at night. We have then to the maximum the non limiting water with the factor,

$$(11) \qquad\qquad ETR_{\max} \text{ ou } ETP_{\max} = (1-a)\,Rg' + Q_{4.5}.$$

This maximum value of ETR or ETP under these conditions could not thus even be exceeded when ETR = 0. We have thus in this case,

$$(12) \qquad ETP \leqslant (1-a)\,Rg' + Q_{4.5}$$

where considering (5),

$$(13) \qquad 2\,ETP_0 \leqslant (1-a)\,Rg + Q_{4.5}$$

which gives,

$$(14) \qquad ETP_0 \leqslant 0.5\,[(1-a)\,Rg + Q_{4.5}].$$

We also deduce,

$$(15) \qquad ETP \leqslant g$$

or

$$(16) \qquad ETP \leqslant a\,\sigma''\,T^4 - (1-a')\,Ra + Qs + Q_3.$$

The potential evapotranspiration can thus be expressed according to the radiative assessment of the long wave ($\sigma T^4$, R'a) in the measurement where the term (Q3) is not too large over a period of 24 hours. The equation (12) permits to understand how it is possible to relate ETP for a given place and certain duration of the day to the nychthemeral amplitude of temperature(the maximal temperature --- the minimal temperature) which is in relation to these exchanges of radiation of the long wave during the cooling phase of the night.

Finally, the equality ETR + ETP = 2 ETP0 is put in the form,

$$(15) \qquad ETR + ETP \leqslant (1-a)\,Rg + Q_{4.5}.$$

In addition, if we indicate by ε the ratio ETR/ETP,

$$(16) \qquad \varepsilon = \frac{ETR}{ETP}$$

εhas the meaning of an index of the relative evapotranspiration equal to 1 for the areas where ETR = ETP and equal to 0 for the desert areas.

The equation (13) permits then to express respectively ETP and ETR.

$$(15) \qquad ETP \leqslant \frac{(1-a)\,Rg + Q_{4.5}}{1+\varepsilon}$$

$$(16) \qquad ETR \leqslant \frac{\varepsilon[(1-a)\,Rg + Q_{4.5}]}{1+\varepsilon}$$

DISCUSSION

The potential evapotranspiration can thus be evaluated in two different ways over multiple
periods of 24 hours when there are no important changes of temperature,
-- or from the radiation of long wave    $(\sigma T^4, Ra)$, which integrates via the temperature the
oasis effects of great scale
--- or from the total absorptive radiation $(1-a)Rg$, the oasis effects of great scale $(Q_4, Q_5)$
and of an index $\varepsilon$ of the relative evapotranspiration.
ETP can not thus be defined only according to the energy factors independently from the
water factor. We will study two limited cases:
When $\varepsilon = 1$

$$ETR = ETP \leqslant 0{,}5[(1-a)\,Rg + Q_{4.5}].$$

When $\varepsilon = 0$    ETR $= 0$

$$ETP \leqslant (1-a)\,Rg + Q_{4.5}.$$

[Figure]

Fig. 2 – The possible maximal variation of ETR and ETP according to $\varepsilon = ETR/ETP$, the
exchange of heating of great scale $Q_{4.5}$ being null

The potential evapotranspiration thus varies to the maximum between 2 limiting values
from *2 ETPo = (1 - a) Rg + Q4.5*    to ETP0 = 0,5[(1 -a)Rg+ Q4.5]    when ETR varies from to ETP. The figure 2 gives the variation of ETR and ETP according to εwhen the oasis effects of great scale are negligible.

Thus, for the equatorial zones where we can admit ε = 1, ETP should be inferior or equal to (1 - a) Rg. Hydrous assessments of some river basins provided by L. TURKISH emphasize an annual ETP about half of the radiation total suitable for be absorbed by an abundant and wet foliage. The hydrous assessments of some river basins provided by L. TURC emphasize an annual ETP in order of the half of the total radiation likely to be absorbed by an abundant and wet foliage.

The same conclusion would be valid for the very large stretches of water such as the seas. However, in the vicinity of the coasts, we will have to take into account of the disturbances introduced by the "calorific wheels" different from the ground and the sea which systematically produce the oasis effects in the form of breeze of sea and of ground of scale equal to or higher than e3. These side exchanges are still increasing with the vicinity of the desert coasts.

If we consider a zone strongly sprinkled such as a very vast oasis in a desert, we can admit that on the edge, we are under the conditions of the desert climate ETR = 0, whereas in center ETR is equal to ETP. Thus from the edge to the center, we can justify from the preceding equations a variation which is from simple to double, all other conditions remaining equal, according to the importance of the guard ring placed around to standardize the conditions. If the preceding inequalities do not give the variation of ETP according to the distance of the considered perimeter, they make it possible to define the higher and lower limits and to find by a very different way, the curves of SUTTON, taken again by CALDER, DUFFEL and LATTAN on the reduction of ETP according to the guard ring (Fig. 3).

[Figure]

Fig. 3 – The variation of ETP according to the ratio L/l from the radiation of the guard ring to that of the measured zone giving ETP.

It will be noted that in an area of mixed-farming, the ETR are very different from each other. This result that ETP will be itself different; but the oasis effects will have, for the effect trending to standardize them, certain zones used as the relatively hot sources, others as the relatively cold sources. According to the importance of the surfaces, the oasis effects will raise different scales. ETP will not only correspond to the average value at the scale superior to that of the turbulence which will not translate ETP of the more reduced surfaces, since it will correspond to the average ETR of the same zone and not ETR of each field.

The preceding whole of the equations supposes that when we pass from ETP to ETR, the transformation is done without perturbing the previous climate. This reasoning is valid only at the limit and will all the more far from the reality, because the ratio between ETR and ETP will vary quickly in the time, or its uniformity zone will possess the dimensions far from the ones corresponding to the scale that defines the climate. Nevertheless, if we consider an uniform zone sufficiently extensive, they allow to define the limits of possible variation of ETR or ETP all linking to the different terms of the assessment *(Rg,* Rnet of the long wave).

Note that the water provision of irrigation has an effect on lowering ETP while highing ETR. This double action contributes to improve strongly the vegetable production. This lowering of ETP, all other conditions remaining equal, will be all the more marked since the treated surface will be important. However, we note that to consider, in a very dry region, ETR as the neighbor of ETP, it will be necessary to irrigate the surface having a very high scale superior to e2. It's thus about a surface corresponding to several $Km^2$ (several hundreds of hectares). In this case, it will be possible to lower strongly ETP and when the scale of the surfaces grows, one will be able to consider the limit, and to arrive at dividing ETP by two. This decrease of ETP for the high scale corresponds not only to a reduction of water consumption, but also to an important improvement of its efficiency.

CONCLUSION

The study of the energy assessment of an uniform region (ground, vegetable cover, nutrition in water) done during a period of 24 hours, allows to establish the simple relations between the real and potential évapotranspiration and the terms of the energy assessment *(Rg, Rnet of the long wave)*. These relations were established with a series of hypotheses that we do not meet generally in the natural conditions; they provide, however, an approach of ETR and of ETP and emphasis the role played by the totally absorbed radiation. The existing compensation between ETP and ETR translated by simple relation ETP + ETR equal to constant for a certain totally absorbed radiation, allows to explain the variations of ETP from these of ETR. We then can have an indication on the order of magnitude of the climatic modifications that we can await of a change of the vegetable cover or of a water provision by carrying out the irrigation to a sufficient scale.

**TABLE 2**

Real Evapotranspiration (ETR) in the Equatorial zone

I --- observed values --- Extract of the thesis of L. TURC "The water assessment of the grounds --- relations amongthe precipitations, the evaporation and the flow"

| Current water, with eventually the surface of the pouring basin and the period of doing the report | Rain in mm | 0°C | ETR |
|---|---|---|---|
| a) Java
Tji Anten (240 km2) 17 years
Tji Kapundung | 4.935 mm
2.650 mm | 21°
18° | 1.188 mm
1.070 mm |
| b) South America
Amazone (report not   known) | 1.900 mm | 24° 5 | 1.245 mm |
| c) Africa
Congo (report not   known)
Sanaga à Edea | 1.400 mm
1.610 mm | 22° 5
23° | 1.030 mm
1.095 mm |

II   ---   Maximum theoretical values of annual ETR

| Total Radiation (Annual average expressed in cal /cm2 /day) | Albedo | |
|---|---|---|
| | 5% | 10% |
| 350 | 1.040 mm | 990 mm |
| 400 | 1.190 mm | 1.130 mm |
| 450 | 1.340 mm | 1.270 mm |

The theoretical maximum values of annual ETR were calculated from the equality *ETR = ETP = ETP0 = 0,5 (1 ~a)Rg* X 365.
The albedo 5% and 10% can correspond to those from a dense forest to the humid foliage.

---

## Referee Comment (RC2) · Anonymous Referee #2 · 25 Aug 2020

Ln 9. Suggest change "Energy correction methods" to "energy balance closure methods" Ln 154-157, does this mean that the two model parameters (i.e. m and n) are determined from alpha and b only? Ln171-177, What is the justification for the treatment of parameter alpha? Was the optimization done for each flux site at daily, weekly, monthly, and annul time scales respectively? Why was equation (5) was tested instead of (6)? Brutsaert (2015) suggested that "it is preferable to use equation (6) and the c parameter should only be introduced to accommodate unusual situations." Ln220-227, The results shown in Figure 1 do not indicate the model performance at daily, weekly, monthly, and annual time scales. If the authors want to know how the model perform at these time scales, they need to show daily to annual results for each site and present a summary of the 88 flux sites. Ln 241, Morton (1983) suggested that the complementary relationship should be applied at longer time scales (e.g. monthly), but it does not explain why the weekly or monthly results are better than the daily results. Ln370, Figure 5 should be Figure 7. What is the significance of this relationship? Ln 409 – 436, This section deals with the issue of the energy balance closure. To me, this is a separate question and I don't see the relevance to the performance of the complementary relationships.

---

## Author Comment (AC2) · 7 Sep 2020

**Response to interactive comment on "At which time scale does the complementary principle perform best on evaporation estimation?" by Liming Wang et al. #2**

(Reviewers comments in Italic and responses in upright Roman)

**Anonymous Referee #2**

General response: Thank you for the timely review. We are very appreciative of your valuable and constructive revision suggestions. The point-by-point responses were provided as follows.

[Figure]

*-Ln 9. Suggest change "Energy correction methods" to "energy balance closure methods"*

Response: Thanks for your advice. The manuscript will be revised accordingly.

*-Ln 154-157, does this mean that the two model parameters (i.e. m and n) are determined from alpha and b only?*

Response: Yes, it is. The variable $x_{0.5}$ in Eq.(4) is also determined by $\alpha$ and b only. Thus, all the parameters in Eq.(4) can be determined from $\alpha$ and b only. Thank you.

*-Ln171-177, What is the justification for the treatment of parameter alpha?*

Response: Thanks for your question. Typically, $\alpha$ has a default value of 1.26 (Priestley Taylor, 1972). Since some studies showed that a constant $\alpha$ may cause irrational results and biases in estimating *E*, it is suggested to specify $\alpha$ for diverse scenarios (Hobbins, Ramírez, Brown, Claessens, 2001b; Ma et al., 2015a; Sugita et al., 2001; Szilagyi, 2007). According to the complementary principle, in wet condition, *E* is close to $E_{pen}$ (Penman evaporation) and the Priestley-Taylor's evaporation ($E_{PT} = \alpha\,E_{rad}$). Specifically, when $E/E_{pen}$ is larger than a threshold (0.9 is commonly adopted), $E_{PT}$ can be considered to approximately equal to the observed *E*, thus $\alpha$ can be calculated by $E/E_{rad}$ (Kahler and Brutsaert, 2006; Ma et al., 2015a). In this study, $\alpha$ was calculated by this method based on the mean value of $E/E_{rad}$ in the wet condition ($E/E_{pen}$ > 0.9). When all the $E/E_{pen}$ values are less than 0.9, $\alpha$ is set as the default value of 1.26. The manuscript will be revised accordingly.

*-Was the optimization done for each flux site at daily, weekly, monthly, and annual time scales respectively?*

Response: Yes, the optimizations were done separately. Thank you.

*-Why was equation (5) was tested instead of (6)? Brutsaert (2015) suggested that "it is preferable to use Eq.(6) and the c parameter should only be introduced to accommodate unusual situations."*

Response: Thanks for your question. Brutsaert (2015) suggested that $c$ should be 0 in usual situations, thus, the PGC function (Eq.(5)) becomes a concise cubic polynomial function including only two terms (Eq.(6)). Although the concise version of the PGC function has been frequently used recently (Brutsaert et al., 2017; Hu et al., 2018; Liu et al., 2016; Zhang et al., 2017), researchers still have different opinions on the true value of $c$. For example, Han and Tian (2018) found that the mean $c$ value of the 20 sites of FLUXNET is $-1$ and Szilagyi et al. (2016) suggested that $c$ is equal to 2 for 334 catchments in America. The results of Zhou et al. (2020) showed that the mean $c$ value is 6.62 for 15 catchments on the Loess Plateau, China. Moreover, we had tested Eq.(6) in the analysis before, and the results showed that the performance of Eq.(6) is much worse than Eq.(5). We provided the results in Table R1 and Figure R1. Since we have used the optimization algorithm to determine the parameter $b$ in the SGC function, it is a fair manner to use the optimal $c$ value instead of a constant value ($c = 0$) in the PGC function. The manuscript will be revised accordingly.

Table R1. The evaluation merits (NSE, $R^2$, and RMSE in W m$^{-2}$) based on Eq.(5) (optimal $c$) with the subscript B-5 and Eq.(6) ($c = 0$) with the subscript B-6.

|  | Day | Week | Month | Year |
|---|---|---|---|---|
| NSE$_{B-5}$ | 0.19 | 0.3 | 0.5 | 0.25 |
| NSE$_{B-6}$ | -0.47 | -0.61 | -0.69 | -8.98 |
| R$^2_{B-5}$ | 0.61 | 0.7 | 0.75 | 0.63 |
| R$^2_{B-6}$ | 0.61 | 0.69 | 0.72 | 0.62 |
| RMSE$_{B-5}$ | 26.83 | 19.17 | 13.7 | 6.96 |
| RMSE$_{B-6}$ | 33.65 | 28.51 | 23.98 | 21.47 |

**Table 1.**

Figure R1. The estimated evaporation based on the polynomial function with $c = 0$ (Eq.(6)) vs the observed evaporation at daily scale (a), weekly scale (b), monthly scale (c), and yearly scale (d).

*-Ln220-227, The results shown in Figure 1 do not indicate the model performance at daily, weekly, monthly, and annual time scales. If the authors want to know how the model performs at these time scales, they need to show daily to annual results for each site and present a summary of the 88 flux sites.*

Response: Thanks for your suggestion. Figure 1 just provides a general cognition of the performance. To accurately show the model efficiency at different time scales, we will provide the results at different timescales for each site in Table S2 following the advice of the reviewer. A summary of these results will be added in the revision.

*-Ln 241, Morton (1983) suggested that the complementary relationship should be applied at longer time scales (e.g. monthly), but it does not explain why the weekly or monthly results are better than the daily results.*

Response: Yes, we agree with the reviewer. Morton (1983) just inferred that the complementary relationship should not be applied at short time scales because of the potential lag times associated with heat and water vapor change (p24 - p25 in Morton, 1983). However, it does not provide solid evidence or theoretical derivation to prove this inference. The statement will be revised. Thank you.

*-Ln370, Figure 5 should be Figure 7. What is the significance of this relationship?*

Response: Thanks for your careful review. The manuscript will be revised accordingly. The relationship provides the additional evidence besides Figure 2 that the two functions can substitute each other in a sense. In other words, the two functions with calibrated parameters substantially provide the similar descriptions of the distribution of results in the state space ( $x = E_{rad}/E_{pen}$, $y = E/E_{pen}$). They can covert to each other in most situations since the two functions are roughly equivalent to the linear asymmetric function when $x$ is neither excessively large nor excessively small.

*-Ln 409 – 436, This section deals with the issue of the energy balance closure. To me, this is a separate question and I don't see the relevance to the performance of the*

*complementary relationships.*

Response: Thanks for your comment. This part will be deleted in the revision.

**References**

Brutsaert, W.: A generalized complementary principle with physical constraints for land-surface evaporation. Water Resour. Res., 51(10), 8087-8093, 2015. https://doi.org/10.1002/2015wr017720

Brutsaert, W., Li, W., Takahashi, A., Hiyama, T., Zhang, L., Liu, W. Z.: Nonlinear advection-aridity method for landscape evaporation and its application during the growing season in the southern Loess Plateau of the Yellow River basin. Water Resour. Res., 53(1), 270-282, 2017. https://doi.org/ 10.1002/2016wr019472

Han, S. J., Tian, F. Q.: Derivation of a sigmoid generalized complementary function for evaporation with physical constraints. Water Resour. Res., 54(7), 5050-5068, 2018. https://doi.org/10.1029/2017wr021755

Hobbins, M. T., Ramirez, J. A., Brown, T. C.: The complementary relationship in estimation of regional evapotranspiration: An enhanced Advection-Aridity model. Water Resour. Res., 37(5), 1389-1403, 2001. https://doi.org/10.1029/2000wr900359

Hu, Z. Y., Wang, G. X., Sun, X. Y., Zhu, M. Z., Song, C. L., Huang, K. W., Chen, X. P.: Spatial-temporal patterns of evapotranspiration along an elevation gradient on Mount Gongga, Southwest China. Water Resour. Res., 54(6), 4180-4192, 2018. https://doi.org/10.1029/2018wr022645

Kahler, D. M., Brutsaert, W.: Complementary relationship between daily evaporation in the environment and pan evaporation. Water Resour. Res, 42(5), 2006. https://doi.org/10.1029/2005WR004541

Liu, X. M., Liu, C. M., Brutsaert, W.: Regional evaporation estimates in the eastern monsoon region of China: Assessment of a nonlinear formulation of the complementary principle. Water Resour. Res., 52(12), 9511-9521, 2016. https://doi.org/10.1002/2016wr019340

Ma, N., Zhang, Y. S., Szilagyi, J., Guo, Y. H., Zhai, J. Q., Gao, H. F.: Evaluating the complementary relationship of evapotranspiration in the alpine steppe of the Tibetan Plateau. Water Resour. Res., 51(2), 1069-1083, 2015a. https://doi.org/10.1002/2014wr015493

Morton, F. I.: Operational estimates of areal evapo-transpiration and their significance to the science and practice of hydrology. J. Hydrol., 66(1-4), 1-76, 1983. https://doi.org/10.1016/0022-1694(83)90177-4

Priestley, C. H. B., Taylor, R. J.: On the assessment of surface heat-flux and evaporation using large-scale parameters. Mon. Weather Rev., 100(2), 81-92, 1972. https://doi.org/10.1175/1520-0493(1972)100<0081:Otaosh>2.3.Co;2

Sugita, M., Usui, J., Tamagawa, I., Kaihotsu, I.: Complementary relationship with a convective boundary layer model to estimate regional evaporation. Water Resour. Res., 37(2), 353-365, 2001. https://doi.org/10.1029/2000wr900299

Szilagyi, J.: On the inherent asymmetric nature of the complementary relationship of evaporation. Geophys. Res. Lett., 34(2), L02405, 1-6, 2007. https://doi.org/10.1029/2006gl028708

Szilagyi, J., Crago, R., Qualls, R. J.: Testing the generalized complementary relationship of evaporation with continental-scale long-term water-balance data. J. Hydrol., 540, 914-922, 2016. https://doi.org/10.1016/j.jhydrol.2016.07.001

Zhang, L., Cheng, L., Brutsaert, W.: Estimation of land surface evaporation using a generalized nonlinear complementary relationship. J. Geophys. Res. Atmos., 122(3), 1475-1487, 2017. https://doi.org/10.1002/2016jd025936

Zhou, H., Han, S., Liu, W.: Evaluation of two generalized complementary functions for annual evaporation estimation on the Loess Plateau, China. J. Hydrol., 124980, 2020. https://doi.org/10.1016/j.jhydrol.2020.124980

Please also note the supplement to this comment:
https://hess.copernicus.org/preprints/hess-2020-379/hess-2020-379-AC2-supplement.pdf

[Figure]

(a)

Day

$y = 0.53 x + 50.21$
$R^2 = 0.23$

$E_{est}$ (W m-2-2)

$E$ (W m$^{-2}$)

(b)

Week

$y = 0.54 x + 39.84$
$R^2 = 0.18$

$E_{est}$ (W m-2-2)

$E$ (W m$^{-2}$)

(c)

Month

$y = 0.55 x + 33.65$
$R^2 = 0.21$

$E_{est}$ (W m-2-2)

$E$ (W m$^{-2}$)

(d)

Year

$y = 0.55 x + 29.03$
$R^2 = 0.23$

$E_{est}$ (W m-2-2)

$E$ (W m$^{-2}$)

**Fig. 1.** Figure R1 The estimated evaporation based on the polynomial function with c = 0 (equation (6)) vs the observed evaporation at daily scale (a), weekly scale (b), monthly scale (c), and yearly scale (d)

[Figure]

---

## Referee Comment (RC3) · Anonymous Referee #3 · 15 Sep 2020

Complementary evaporation relationships have been studied at multiple time scales, which time scale is the most suitable one? In this respect, the manuscript gave very meaningful results. It is recommended that the draft should be revised on the following questions before publication.

(1). Ln172-173, Ln458-459, "When all the E/Epen values were less than 0.9, alpha was set as the default value of 1.26". This default value is problematic for the PGC model. The independent variable of PGC model is Epo/Epa = alpha*Erad/Epen, which is less than or equal to 1. When alpha =1.26, the range of Erad/Epen values is only 0-0.79. However, if alpha =1, the range of Erad/Epen values is 0-1. It could be imagined that the PGC can not fit the data points with 0.79<Erad/Epen<1 if the alpha =1.26, but there is no problem in the case of alpha =1.

[Figure]

(2). Ln294-295, Ln336-337, Ln351-352,Ln466-467, The manuscript gave a conclusion that the parameter c of PGC model decreased with the increase of time scale. The parameter c was determined under the condition of a fixed alpha in this study, which needs to be specially explained. When the c is a fixed value, say 0, the alpha would change with the month (Liu et al.,2016).

(3). By using statistical indexes such as determination coefficient, the manuscript considered that the complementary relationship of a monthly scale was the best, but the other time scales were not poor and reached to a very significant level too. Does this mean that the complementary relationship on other time scales also exists significantly, not as Morton (1983) said, only at longer timescales?

(4). Ln23, "globle water and energy cycle". Generally, water can have a cycle, but energy flows only.

References Liu, X., C. Liu W. Brutsaert.2016. Regional evaporation estimates in the eastern monsoon region of China: Assessment of a nonlinear formulation of the complementary principle. Water Resources Research, 52: 9511-9521. Morton, F. I. 1983. Operational estimates of areal evapotranspiration and their significance to the science and practice of hydrology. Journal of Hydrology, 66(1-4):1-76.

---

## Author Comment (AC3) · 7 Oct 2020

**Response to interactive comment on "At which time scale does the complementary principle perform best on evaporation estimation?" by Liming Wang et al. #3**

(Reviewers comments in Italic and responses in upright Roman)

**Anonymous Referee #3**

*Complementary evaporation relationships have been studied at multiple time scales, which time scale is the most suitable one? In this respect, the manuscript gave very meaningful results. It is recommended that the draft should be revised on the following questions before publication.*

[Figure]

General response: Thanks for your careful review and affirmation of this work. All the questions are very constructive and inspiring. The point-by-point responses were provided as follows.

*-(1). Ln172-173, Ln458-459, "When all the $E/E_{pen}$ values were less than 0.9, $\alpha$ was set as the default value of 1.26". This default value is problematic for the PGC model. The independent variable of PGC model is $E_{po}/E_{pa} = \alpha * E_{rad}/E_{pen}$, which is less than or equal to 1. When $\alpha$ =1.26, the range of $E_{rad}/E_{pen}$ values is only 0-0.79. However, if $\alpha$ =1, the range of $E_{rad}/E_{pen}$ values is 0-1. It could be imagined that the PGC cannot fit the data points with $0.79<E_{rad}/E_{pen}<1$ if the $\alpha$ =1.26, but there is no problem in the case of $\alpha$ =1.*

Response: Thanks for your comment. Indeed, the PGC model does not work for the range of $0.79 < E_{rad}/E_{pen} < 1.0$ when $\alpha$ adopts its default value of 1.26 (Priestley Taylor, 1972; Brutsaert Stricker, 1979), which is a shortage of PGC. In our manuscript, $\alpha$ was calculated by the mean value of the ratio of $E_{PT}$ to $E_{rad}$ during the study period (similar treatment can be found in Kahler Brutsaert, 2006). Such calculation is based on the physical definition of the Priestley-Taylor coefficient (i.e., $\alpha$). Actually, the values of $\alpha$ for all sites besides those adopting $\alpha$ = 1.26 are greater than 1.0 in our study, which means the PGC model cannot work properly for the condition of $1/\alpha < E_{rad}/E_{pen} <1.0$.

In the submitted manuscript, the original results for $1/\alpha < E_{rad}/E_{pen} < 1$ calculated by the PGC function were kept. We have carried out an additional analysis that adopting $E = E_{pen}$ for $1/\alpha < E_{rad}/E_{pen} < 1$ in the PGC function, and the resultant NSE$_B$ (0.19 vs 0.19) and RMSE$_B$ (26.83 W m$^{-2}$ vs 26.68 W m$^{-2}$) presented very similar results. The manuscript will be revised to incorporate these discussions. Thank you.

*-(2). Ln294-295, Ln336-337, Ln351-352,Ln466-467, The manuscript gave a conclusion that the parameter c of PGC model decreased with the increase of time scale. The parameter c was determined under the condition of a fixed $\alpha$ in this study, which needs*

*to be specially explained. When the c is a fixed value, say 0, the $\alpha$ would change with the month (Liu et al.,2016).*

Response: Thanks for your comment. To make the model parsimonious, it is a reasonable choice to give one value for the parameters $\alpha$ and *c* at each site for every different time scale. If the parameter was alterable, for example, it was monthly dependent, we will have to calibrate 12 parameters instead of one value for the whole study period. The purpose of this study is to find the most suitable timescale for the complementary functions, the variances of the key parameter within a timescale will introduce extra uncertainties. It is true that the accuracy will increase when an alterable parameter (that means higher number of parameters) is used, however, the probability of overfitting risk will increase at the same time. Besides, a general representation of the parameter is more helpful to detect its overall trend as the change of timescale than a group of parameters.

Moreover, we carried out an additional analysis that *c* is fixed to 0, and $\alpha$ is calibrated as $\alpha_e$. We found that the two methods gave similar results (mean RMSE = 14.99 W m$^{-2}$ for $\alpha_e$ vs 16.67 W m$^{-2}$ for $\alpha$) and the conclusion on the time scale issue is consistent by adopting either $\alpha$ or $\alpha_e$ in the analysis. Actually, the optimal $\alpha_e$ has a significantly negative linear relationship with the optimal *c* and the Pearson correlation coefficient is $-0.8$. It suggests that calibrating either of the two parameters ($\alpha_e$ and *c*) equivalent (Han et al., 2012). Thanks all the same, and the manuscript will be revised accordingly to incorporate these discussions.

*-(3). By using statistical indexes such as determination coefficient, the manuscript considered that the complementary relationship of a monthly scale was the best, but the other time scales were not poor and reached to a very significant level too. Does this mean that the complementary relationship on other time scales also exists significantly, not as Morton (1983) said, only at longer timescales?*

Response: Thanks for your question. Yes, we found the two complementary functions perform reasonably well at shorter timescales (i.e., day and week) with pretty high $R^2$. Also, the estimations of site mean evaporation at shorter timescales are accurate (Figure 1 and Figure 3), especially for the SGC function. These indeed suggest the complementary relationship holds at relatively shorter time scales, or at least we can say that the generalized complementary functions have the ability to estimate the evaporation accurately even at the shorter timescales. The manuscript will be revised to incorporate these discussions. Thanks.

*-(4). Ln23, "global water and energy cycle". Generally, water can have a cycle, but energy flows only.*

Response: Thanks for your careful review. The statement will be revised as "global water cycle and energy balance".

**References**

Brutsaert, W., Stricker, H.: Advection-Aridity approach to estimate actual regional evapotranspiration. Water Resour. Res., 15(2), 443-450, 1979. https://doi.org/10.1029/WR015i002p00443

Han, S. J., Hu, H. P., Tian, F. Q.: A nonlinear function approach for the normalized complementary relationship evaporation model. Hydrol. Processes, 26(26), 3973-3981, 2012. https://doi.org/10.1002/hyp.8414

Kahler, D. M., Brutsaert, W.: Complementary relationship between daily evaporation in the environment and pan evaporation. Water Resour. Res, 42(5), 2006. https://doi.org/10.1029/2005WR004541

Liu, X. M., Liu, C. M., Brutsaert, W.: Regional evaporation estimates in the eastern monsoon region of China: Assessment of a nonlinear formulation of the complementary principle. Water Resour. Res., 52(12), 9511-9521, 2016. https://doi.org/10.1002/2016wr019340

Morton, F. I.: Operational estimates of areal evapo-transpiration and their significance to the science and practice of hydrology. J. Hydrol., 66(1-4), 1-76, 1983.

https://doi.org/10.1016/0022-1694(83)90177-4

Priestley, C. H. B., Taylor, R. J.: On the assessment of surface heat-flux and evaporation using large-scale parameters. Mon. Weather Rev., 100(2), 81-92, 1972. https://doi.org/10.1175/1520-0493(1972)100<0081:Otaosh>2.3.Co;2

Please also note the supplement to this comment:
https://hess.copernicus.org/preprints/hess-2020-379/hess-2020-379-AC3-supplement.pdf

---

## Author Response (AR1)

| 3      |                                                                                                                                                                                       |
| 4      | (Reviewers comments in Italic and responses in upright Roman)                                                                                                                         |
| 5      |                                                                                                                                                                                       |
| 6      | Editor                                                                                                                                                                                |
| 7      |                                                                                                                                                                                       |
| 8
9 | Your paper has been reviewed by 3 reviewers. You have received 1 very critical review, and 2 reviews that consider proceeding with revisions. You've provided detailed replies to the |
| 10     | concern as raised by the reviewers. After carefully reading the review reports and your                                                                                               |
| 11     | rebuttal. I suggest to proceed with a major revision in which you include the suggested                                                                                               |
| 12     | revisions. The revised manuscript shall be resend to the critical referee for a review of the                                                                                         |
| 13     | revised manuscript                                                                                                                                                                    |
| 14     |                                                                                                                                                                                       |
| 15     | Thank you for providing the opportunity to submit the revision of this work. We sincerely                                                                                             |
| 16     | appreciate the comments of both editor and the reviewers. The detailed constructive                                                                                                   |
| 17     | comments are critically important for us to improve the manuscript. In response to these                                                                                              |
| 18     | comments, we have performed additional analyses and substantially improved the quality of                                                                                             |
| 19     | the manuscript. We hope that the editor and reviewers will be favorably impressed by the                                                                                              |
| 20     | revised version. The point-by-point responses were provided as follows.                                                                                                               |
| 21     |                                                                                                                                                                                       |
| 22     | Anonymous Referee #1                                                                                                                                                                  |
| 23     | ·                                                                                                                                                                                     |
| 24     | -The MS is carelessly written. It should be thoroughly rechecked for grammar, typos,                                                                                                  |
| 25     | language constructs.                                                                                                                                                                  |
| 26     |                                                                                                                                                                                       |
| 27     | Response: Thank you for the comments. We have gone through and revise the manuscript                                                                                                  |
| 28     | thoroughly and hired some language experts to help polish the manuscript again.                                                                                                       |
| 29     |                                                                                                                                                                                       |
| 30     | -For example, the AA method is mentioned several times before it is explained.                                                                                                        |
| 31     |                                                                                                                                                                                       |
| 32     | Response: Thank you for pointing out this problem. we had provided the full name for it                                                                                               |
| 33     | when the first time it is mentioned (Line 55-57, hereafter all lines numbers are based on                                                                                             |
| 34     | the tracked version). Also, we moved the explanation from the methodology part to the                                                                                                 |
| 35     | introduction part.                                                                                                                                                                    |
| 36     |                                                                                                                                                                                       |
| 37     | -Also, the first asymmetric AA method was of Kahler and Brutsaert (2006), and not by                                                                                                  |
| 38     | Brutsaert and Parlange (1998).                                                                                                                                                        |
| 39     |                                                                                                                                                                                       |
| 40     | Response: According to our reading, Brutsaert and Parlange (1998) provided the following                                                                                              |
| 41     | equation in their paper:                                                                                                                                                              |
| 42     | $E = \left[ (1+b)E_0 - aE_{pa} \right] / b$                                                                                                                                           |

| 43
44 | where, $E_0$ has the same meaning of $E_{po}$ in our manuscript (i.e., potential evaporation), and a is
a pan coefficient, b is an asymmetric parameter. Our statement "the CR was extended to a |
|----------|-------------------------------------------------------------------------------------------------------------------------------------------------------------------------------------------------------------------|
| 45       | linear function with an asymmetric parameter (Brutsaert and Parlange, 1998)" (Line 57-58)                                                                                                                         |
| 46       | refers to this equation.                                                                                                                                                                                          |
| 47       | 1                                                                                                                                                                                                                 |
| 48
49 | Kahler and Brutsaert (2006) summarized the previous work of Brutsaert and Stricker (1979),
Brutsaert and Parlange (1998), and Brutsaert (2005) and gave the equation:                                          |
| 50       |                                                                                                                                                                                                                   |
| 51       | $(1+b)E_0 = C_p E_{pa} + bE$                                                                                                                                                                                      |
| 52       | where, $C_p$ is a constant parameter. We can see that this equation holds the same format with                                                                                                                    |
| 53       | Brutsaert and Parlange (1998) after appropriate transformation (and replacing $C_{\rm P}$ with a). It                                                                                                             |
| 54       | may be the first time it was called "asymmetric AA". Thank you.                                                                                                                                                   |
| 55       |                                                                                                                                                                                                                   |
| 56       | -Also, nobody reads the original work of Bouchet (1963), it seems, as it is in French. That                                                                                                                       |
| 57       | may be the reason for frequent misquoting it. My understanding is that he never proposed a                                                                                                                        |
| 58       | symmetrical CR. Even Brutsaert in his seminal book (1982) is controversial about this issue.                                                                                                                      |
| 59       | The authors should clarify this issue though.                                                                                                                                                                     |
| 60       |                                                                                                                                                                                                                   |
| 61       | Response: Yes, the original work of Bouchet (1963) is French. In our institute of Tsinghua                                                                                                                        |
| 62       | University, we have a PhD student coming from France, and he had translated this paper into                                                                                                                       |
| 63       | English several years ago. We are pleased to provide the English version of Bouchet (1963)                                                                                                                        |
| 64       | at the end of the response for the reference. After reading this paper, we suggest that the                                                                                                                       |
| 65       | contribution of Bouchet (1963) should be respected                                                                                                                                                                |
| 66       |                                                                                                                                                                                                                   |
| 67       | Equation (5) and Figure 2 of Bouchet (1963) show a symmetrical complementary                                                                                                                                      |
| 68       | relationshin:                                                                                                                                                                                                     |
| 69       | $ETP + ETR = 2ETP_{o}$                                                                                                                                                                                            |
| 70       | where ETR is the energy corresponding to the real evapotranspiration ETP is corresponding                                                                                                                         |
| 71       | to $E_{\rm P2}$ and $ETP_0$ is corresponding to $E_{\rm P2}$                                                                                                                                                      |
| 72       | to Dpa, and DTT o is corresponding to Dpo.                                                                                                                                                                        |
| 73       | In the book of Brutsaert (1982, $p224-225$ ), the above equation is cited as equation (10.35)                                                                                                                     |
| 74       | and Brutsaert said that Bouchet (1963) arrived at the complementary relationship and admit                                                                                                                        |
| 75       | Bouchet's approach contains worthwhile ideas and led to further developments. Brutsaert                                                                                                                           |
| 76       | thought this method is not used widely because the assumption is strict and it did not provide                                                                                                                    |
| 77       | exactly measures of $F_{res}$ and $F_{res}$ . Thank you                                                                                                                                                           |
| 78       | exactly measures of D pa and D po . Thank you.                                                                                                                                              |
| 79       | -I do not really see what we gain from this study. The high NSF value for the month comes                                                                                                                         |
| 80
80 | about because its high variance between months and it is already being long enough to                                                                                                                             |
| 81       | smooth things out                                                                                                                                                                                                 |
| 82       | smooth things out.                                                                                                                                                                                                |
| 83       | Response: The aim of this study is to investigate at which time scale the complementary                                                                                                                           |
| 84       | nrinciple performs best on evaporation estimation. Based on this reviewer's comment, we                                                                                                                           |
| 85       | understand that the reviewer gained that complementary functions perform best at the                                                                                                                              |
| 86       | monthly scale. Actually, it's exactly what we want to convey to the audience. We did not find                                                                                                                     |

the evidence in previous studies or theoretical derivation which had already revealed this
conclusion. Without these results, it is still uncertain how long is "enough to smooth things
out". It could be 7 days, 30 days or 90 days. We agree with the reasons for the high NSE
value at the monthly scale given by the reviewer, these reasons are also discussed in our

91 manuscript (Line 268- 272). The "high variance" can be corresponding to our explanation

92 about "variabilities of x and y" (Line 272), and the "smooth things out" can be corresponding

- 93 to our explanation of RMSE. Thank you.
- 94

*-I bet that between Mays, Junes, Julys, etc., the NSE value would not be better than for the seasons and years.*

97

98 Response: In the current version, the study periods are from April to September for the

99 Northern Hemisphere and from October to March for the Southern Hemisphere. We

supposed that the reviewer mean that if the study periods are shortened (e.g, from May to

101 July), the NSE values at the monthly scale will not be better than for the seasons and years.

102 We have provided the results for May to July in **Table R1**. In this situation, the seasonal

result is equal to the annual result and there is one seasonal result (May to July) each year.

104 These results still support our conclusion. The NSE values at the monthly scale (NSEH = 0.38

and NSEB = 0.32) are higher than those at the seasonal/annual scale (NSEB = -0.07 and NSEB = -0.05). Thank you for providing an opportunity to test the uncertainty in the length

- 107 of study periods.
- 108

**Table R1.** The evaluation merits (NSE, R2 and, RMSE in W m-2) of the two generalized
 complementary functions from May to July

|                          | Month | Season/Year |
|--------------------------|-------|-------------|
| NSE H             | 0.38  | -0.07       |
| NSE B  | 0.32  | -0.05       |
| $\mathbf{R}^{2}$ H       | 0.63  | 0.56        |
| $R^{2}B$                 | 0.63  | 0.56        |
| RMSE H | 12.17 | 8.86        |
| RMSE B | 21.51 | 8.81        |

111 112

113 *-The low value for the annual time-scale is a bit worrisome as it means that these two chosen*

114 *methods cannot replicate any long-term trends in ET rates to acceptable accuracy, which*

115 *diminishes their potential values for long-term hydrological modeling.*

116

117 Response: Yes, the complementary functions perform worse in estimating *E* at the annual

scale. To the best of our knowledge, this point had not been thoroughly discussed previously.

119 We did not recommend choosing the annual scale as the timestep to estimate E because of the

120 low efficiency. However, we can still replicate the long-term trends in *E* rates by adopting the

121 monthly timestep. Thank you.

| 123
124               | Anonymous Referee #2                                                                                                                                                                                                                                                                                                                                                                                                                              |
|--------------------------|---------------------------------------------------------------------------------------------------------------------------------------------------------------------------------------------------------------------------------------------------------------------------------------------------------------------------------------------------------------------------------------------------------------------------------------------------|
| 125
126               | -Ln 9. Suggest change "Energy correction methods" to "energy balance closure methods"                                                                                                                                                                                                                                                                                                                                                             |
| 127
128               | Response: Thanks for your advice. The manuscript has been revised accordingly (Line 9).                                                                                                                                                                                                                                                                                                                                                           |
| 129
130
131        | -Ln 154-157, does this mean that the two model parameters (i.e. m and n) are determined from alpha and b only?                                                                                                                                                                                                                                                                                                                                    |
| 132                      | Response: Yes, it is. The variable $x_{0.5}$ in Eq. (4) is also determined by $\alpha$ and $b (x_{0.5} = \frac{0.5+b^{-1}}{\alpha(1+b^{-1})})$                                                                                                                                                                                                                                                                                                    |
| 133
134               | only. Thus, all the parameters in Eq. (4) can be determined from $\alpha$ and b only. Thank you.                                                                                                                                                                                                                                                                                                                                                  |
| 135
136               | -Ln171-177, What is the justification for the treatment of parameter alpha?                                                                                                                                                                                                                                                                                                                                                                       |
| 137
138
139
140 | Response: Thanks for your question. Typically, $\alpha$ has a default value of 1.26 (Priestley & Taylor, 1972). Since some studies showed that a constant $\alpha$ may cause irrational results and biases in estimating E , it is suggested to specify $\alpha$ for diverse scenarios (Hobbins, Ramírez, Brown, & Claessens, 2001; Ma et al., 2015; Sugita et al., 2001; Szilagyi, 2007). According to                                    |
| 141
142
143
144 | the complementary principle, in wet condition, E is close to $E_{pen}$ (Penman evaporation) and
the Priestley-Taylor's evaporation ( $E_{PT} = \alpha E_{rad}$ ). Specifically, when $E/E_{pen}$ is larger than a
threshold (0.9 is commonly adopted), $E_{PT}$ can be considered to approximately equal to the
observed E , thus $\alpha$ can be calculated by $E/E_{rad}$ (Kahler and Brutsaert, 2006; Ma et al., 2015). |
| 145
146
147
148 | In this study, $\alpha$ was calculated by this method based on the mean value of $E/E_{rad}$ in the wet condition ( $E/E_{pen} > 0.9$ ). When all the $E/E_{pen}$ values are less than 0.9, $\alpha$ is set as the default value of 1.26. The manuscript has been revised accordingly (Lines 175-186).                                                                                                                                            |
| 149
150
151        | -Was the optimization done for each flux site at daily, weekly, monthly, and annual time scales respectively?                                                                                                                                                                                                                                                                                                                                     |
| 151
152
153        | Response: Yes, the optimizations were done separately. Thank you.                                                                                                                                                                                                                                                                                                                                                                                 |
| 154
155
156
157 | -Why was equation (5) was tested instead of (6)? Brutsaert (2015) suggested that "it is preferable to use equation (6) and the c parameter should only be introduced to accommodate unusual situations."                                                                                                                                                                                                                                          |
| 157
158
159        | Response: Thanks for your question. Brutsaert (2015) suggested that $c$ should be 0 in usual situations, thus, the PGC function (Eq. (5)) becomes a concise cubic polynomial function                                                                                                                                                                                                                                                             |
| 160
161
162
163 | including only two terms (Eq. (6)). Although the concise version of the PGC function has been frequently used recently (Brutsaert et al., 2017; Hu et al., 2018; Liu et al., 2016; Zhang et al., 2017), researchers still have different opinions on the true value of $c$ . For example, Han and Tian (2018) found that the mean $c$ value of the 20 sites of FLUXNET is $-1$ and Szilagvi                                                       |
| 164
165               | et al. (2016) suggested that $c$ is equal to 2 for 334 catchments in America. The results of Zhou et al. (2020) showed that the mean $c$ value is 6.62 for 15 catchments on the Loess Plateau,                                                                                                                                                                                                                                                    |

166 China. Moreover, we had tested Eq. (6) in the analysis before, and the results showed that the 167 performance of Eq. (6) is much worse than Eq. (5). We provided the results in **Table R2** and 168 **Figure R1**. Since we have used the optimization algorithm to determine the parameter *b* in 169 the SGC function, it is a fair manner to use the optimal *c* value instead of a constant value (*c* 170 = 0) in the PGC function. The manuscript has been revised accordingly (Lines 191-193).

171

**Table R2.** The evaluation merits (NSE,  $\mathbb{R}^2$ , and RMSE in W m-2) based on Eq. (5) (optimal *c*) with the subscript B-5 and Eq. (6) (*c* = 0) with the subscript B-6.

|                      | Day   | Week  | Month | Year  |
|----------------------|-------|-------|-------|-------|
| NSE B-5   | 0.19  | 0.3   | 0.5   | 0.25  |
| NSE B-6   | -0.47 | -0.61 | -0.69 | -8.98 |
| $\mathbb{R}^{2}$ B-5 | 0.61  | 0.7   | 0.75  | 0.63  |
| $\mathbb{R}^{2}$ B-6 | 0.61  | 0.69  | 0.72  | 0.62  |
| RMSE B-5  | 26.83 | 19.17 | 13.7  | 6.96  |
| RMSE B-6  | 33.65 | 28.51 | 23.98 | 21.47 |

174

175

**Figure R1.** The estimated evaporation based on the polynomial function with c = 0 (equation (6)) vs the observed evaporation at daily scale (a), weekly scale (b), monthly scale (c), and yearly scale (d).

179

181 weekly, monthly, and annual time scales. If the authors want to know how the model performs

- at these time scales, they need to show daily to annual results for each site and present a
  summary of the 88 flux sites.
- **185** Response: Thanks for your suggestion. Figure 1 just provides a general cognition of the
- performance. To accurately show the model efficiency at different time scales, we have
   provided the results at different timescales for each site in **Table S2** following the advice of
- the reviewer. A summary of these results have been added in the revision (Lines 259-262).
- 189
- -Ln 241, Morton (1983) suggested that the complementary relationship should be applied at
  longer time scales (e.g. monthly), but it does not explain why the weekly or monthly results
  are better than the daily results.
- 193
- 194 Response: Yes, we agree with the reviewer. Morton (1983) just inferred that the
- complementary relationship should not be applied at short time scales because of the
- 196 potential lag times associated with heat and water vapor change (p24 p25 in Morton, 1983).
- 197 However, it does not provide solid evidence or theoretical derivation to prove this inference.
- 198 The statement has been revised (Lines 272-277). Thank you.
- 199
- 200 -Ln370, Figure 5 should be Figure 7. What is the significance of this relationship?
- 201

Response: Thanks for your careful review. The manuscript has been revised accordingly
 (Lines 419-426). The relationship provides the additional evidence besides Figure 2 that the
 two functions can substitute each other in a sense. In other words, the two functions with

205 calibrated parameters substantially provide the similar descriptions of the distribution of

- results in the state space ( $x = E_{rad}/E_{pen}$ ,  $y = E/E_{pen}$ ). They can covert to each other in most
- situations since the two functions are roughly equivalent to the linear asymmetric functionwhen *x* is neither excessively large nor excessively small.
- 209

-Ln 409 – 436, This section deals with the issue of the energy balance closure. To me, this is
a separate question and I don't see the relevance to the performance of the complementary
relationships.

- 213
- 214 Response: Thanks for your comment. This part has been deleted in the revision.
- 215

**216 Anonymous Referee #3**

217

218 *Complementary evaporation relationships have been studied at multiple time scales,*

- 219 which time scale is the most suitable one? In this respect, the manuscript gave very
- 220 meaningful results. It is recommended that the draft should be revised on the following

221 *questions before publication.*

222

Thanks for your careful review and affirmation of this work. All the questions are veryconstructive and inspiring. The point-by-point responses were provided as follows.

225

226 -(1). Ln172-173, Ln458-459, "When all the  $E/E_{pen}$  values were less than 0.9,  $\alpha$  was

set as the default value of 1.26". This default value is problematic for the PGC model.

228 The independent variable of PGC model is  $E_{po}/E_{pa} = \alpha * E_{rad}/E_{pen}$ , which is less

than or equal to 1. When  $\alpha = 1.26$ , the range of  $E_{rad}/E_{pen}$  values is only 0-0.79.

230 *However, if*  $\alpha = 1$ *, the range of*  $E_{rad}/E_{pen}$  *values is* 0-1*. It could be imagined that*

231 the PGC cannot fit the data points with  $0.79 < E_{rad}/E_{pen} < 1$  if the  $\alpha = 1.26$ , but there

232 *is no problem in the case of*  $\alpha = 1$ *.*

233

Response: Thanks for your comment. Indeed, the PGC model does not work for the range of

235  $0.79 < E_{\rm rad}/E_{\rm pen} < 1.0$  when  $\alpha$  adopts its default value of 1.26 (Priestley & Taylor, 1972;

**236** Brutsaert & Stricker, 1979), which is a shortage of PGC. In our manuscript,  $\alpha$  was calculated

by the mean value of the ratio of  $E_{PT}$  to  $E_{rad}$  during the study period (similar treatment can be found in Kahler & Brutsaert, 2006). Such calculation is based on the physical definition of

226 Found in Ramer & Drussent, 2000). Such calculation is based on the physical definition of

- the Priestley-Taylor coefficient (i.e.,  $\alpha$ ). Actually, the values of  $\alpha$  for all sites besides those adopting  $\alpha = 1.26$  are greater than 1.0 in our study, which means the PGC model cannot work properly for the condition of  $1/\alpha < E_{rad}/E_{pen} < 1.0$ .
- 242

In the submitted manuscript, the original results for  $1/\alpha < E_{rad}/E_{pen} < 1$  calculated by the PGC function were kept. We have carried out an additional analysis that adopting  $E = E_{pen}$  for  $1/\alpha$  $< E_{rad}/E_{pen} < 1$  in the PGC function, and the resultant NSEB (0.19 vs 0.19) and RMSEB (26.83 W m-2 vs 26.68 W m-2) presented very similar results. The manuscript has been revised to incorporate these discussions (**Lines 365-367**). Thank you.

248

249 -(2). Ln294-295, Ln336-337, Ln351-352, Ln466-467, The manuscript gave a conclusion

that the parameter c of PGC model decreased with the increase of time scale. The

251 parameter c was determined under the condition of a fixed  $\alpha$  in this study, which

needs to be specially explained. When the c is a fixed value, say 0, the  $\alpha$  would

change with the month (Liu et al., 2016).

254

255 Response: Thanks for your comment. To make the model parsimonious, it is a reasonable

256 choice to give one value for the parameters  $\alpha$  and *c* at each site for every different time scale.

257 If the parameter was alterable, for example, it was monthly dependent, we will have to

calibrate 12 parameters instead of one value for the whole study period. The purpose of this

study is to find the most suitable timescale for the complementary functions, the variances of

the key parameter within a timescale will introduce extra uncertainties. It is true that the
accuracy will increase when an alterable parameter (that means higher number of parameters)
is used, however, the probability of overfitting risk will increase at the same time. Besides, a
general representation of the parameter is more helpful to detect its overall trend as the
change of timescale than a group of parameters.

265

Moreover, we carried out an additional analysis that c is fixed to 0, and  $\alpha$  is calibrated as  $\alpha_{e}$ . 266 We found that the two methods gave similar results (mean RMSE = 14.99 W m-2 for  $\alpha_e$  vs 267 16.67 W m-2 for  $\alpha$ ) and the conclusion on the time scale issue is consistent by adopting either 268  $\alpha$  or  $\alpha_e$  in the analysis. Actually, the optimal  $\alpha_e$  has a significantly negative linear relationship 269 with the optimal c and the Pearson correlation coefficient is -0.8. It suggests that calibrating 270 either of the two parameters ( $\alpha_e$  and c) equivalent (Han et al., 2012). Thanks all the same, and 271 the manuscript has been revised accordingly to incorporate these discussions (Lines 195-204, 272 437-443). 273

274

-(3). By using statistical indexes such as determination coefficient, the manuscript considered
that the complementary relationship of a monthly scale was the best, but the

277 other time scales were not poor and reached to a very significant level too. Does this

278 mean that the complementary relationship on other time scales also exists significantly,

279 not as Morton (1983) said, only at longer timescales?

280

Response: Thanks for your question. Yes, we found the two complementary functions
perform reasonably well at shorter timescales (i.e., day and week) with pretty high R2. Also,
the estimations of site mean evaporation at shorter timescales are accurate (Figure 1 and
Figure 3), especially for the SGC function. These indeed suggest the complementary
relationship holds at relatively shorter time scales, or at least we can say that the generalized

complementary functions have the ability to estimate the evaporation accurately even at the
shorter timescales. The manuscript has been revised to incorporate these discussions (Lines
373-377). Thanks.

289

-(4). Ln23, "global water and energy cycle". Generally, water can have a cycle, but
energy flows only.

292

Response: Thanks for your careful review. The statement has been revised as "global water
cycle and energy balance" (Line 25).

- 295
- 296
- 297 298
- 299
- 300

301

**303 **References**

- Bouchet, R. J.: Evapotranspiration réelle et potentielle, signification climatique. IAHS Publ, 62, 134-142,
  1963.
- 306 Brutsaert, W.: Evaporation into the atmosphere: theory, history and applications. Springer, New York, 1982
- 307 Brutsaert, W.,: Hydrology: An Introduction, 605 pp., Cambridge Univ. Press, New York. 2005
- **308** Brutsaert, W.: A generalized complementary principle with physical constraints for land-surface
- evaporation. Water Resour. Res., 51(10), 8087-8093, 2015. https://doi.org/10.1002/2015wr017720
  Brutsaert, W., Li, W., Takahashi, A., Hiyama, T., Zhang, L., Liu, W. Z.: Nonlinear advection-aridity method
  for landscape evaporation and its application during the growing season in the southern Loess
  Plateau of the Yellow River basin. Water Resour. Res., 53(1), 270-282, 2017. https://doi.org/
- **313** 10.1002/2016wr019472
- Brutsaert, W., Parlange, M. B.: Hydrologic cycle explains the evaporation paradox. Nature, 396(6706), 3030, 1998. https://doi.org/ 10.1038/23845
- Brutsaert, W., Stricker, H.: Advection-Aridity approach to estimate actual regional evapotranspiration.
  Water Resour. Res., 15(2), 443-450, 1979. https://doi.org/ 10.1029/WR015i002p00443
- Han, S. J., Hu, H. P., Tian, F. Q.: A nonlinear function approach for the normalized complementary
  relationship evaporation model. Hydrol. Processes, 26(26), 3973-3981, 2012.
  https://doi.org/10.1002/hyp.8414
- Han, S. J., Tian, F. Q.: Derivation of a sigmoid generalized complementary function for evaporation with
- 322
   physical constraints. Water Resour. Res., 54(7), 5050-5068, 2018.

   323
   https://doi.org/10.1029/2017wr021755
- Hobbins, M. T., Ramirez, J. A., Brown, T. C.: The complementary relationship in estimation of regional
  evapotranspiration: An enhanced Advection-Aridity model. Water Resour. Res., 37(5), 1389-1403,
  2001. https://doi.org/10.1029/2000wr900359
- Hu, Z. Y., Wang, G. X., Sun, X. Y., Zhu, M. Z., Song, C. L., Huang, K. W., Chen, X. P.: Spatial-temporal
  patterns of evapotranspiration along an elevation gradient on Mount Gongga, Southwest China.
  Water Resour. Res., 54(6), 4180-4192, 2018. https://doi.org/10.1029/2018wr022645
- Kahler, D. M., Brutsaert, W.: Complementary relationship between daily evaporation in the environment
   and pan evaporation. Water Resour. Res, 42(5), 2006. https://doi.org/10.1029/2005WR004541
- Liu, X. M., Liu, C. M., Brutsaert, W.: Regional evaporation estimates in the eastern monsoon region of
   China: Assessment of a nonlinear formulation of the complementary principle. Water Resour.
   Res., 52(12), 9511-9521, 2016. https://doi.org/10.1002/2016wr019340
- Ma, N., Zhang, Y. S., Szilagyi, J., Guo, Y. H., Zhai, J. Q., Gao, H. F.: Evaluating the complementary
  relationship of evapotranspiration in the alpine steppe of the Tibetan Plateau. Water Resour. Res.,
  51(2), 1069-1083, 2015. https://doi.org/10.1002/2014wr015493
- Morton, F. I.: Operational estimates of areal evapo-transpiration and their significance to the science and
   practice of hydrology. J. Hydrol., 66(1-4), 1-76, 1983. https://doi.org/10.1016/0022 1694(83)90177-4
- Priestley, C. H. B., Taylor, R. J.: On the assessment of surface heat-flux and evaporation using large-scale
  parameters. Mon. Weather Rev., 100(2), 81-92, 1972. https://doi.org/10.1175/15200493(1972)100<0081:Otaosh>2.3.Co;2
- Sugita, M., Usui, J., Tamagawa, I., & Kaihotsu, I.: Complementary relationship with a convective
   boundary layer model to estimate regional evaporation. Water Resour. Res., 37(2), 353-365, 2001.

https://doi.org/10.1029/2000wr900299 346 Szilagyi, J.: On the inherent asymmetric nature of the complementary relationship of evaporation. 347 Geophys. Res. Lett., 34(2), L02405, 1-6, 2007. https://doi.org/10.1029/2006gl028708 348 Szilagyi, J., Crago, R., & Qualls, R. J.: Testing the generalized complementary relationship of evaporation 349 with continental-scale long-term water-balance data. J. Hydrol., 540, 914-922, 2016. 350 351 https://doi.org/10.1016/j.jhydrol.2016.07.001 Zhang, L., Cheng, L., Brutsaert, W.: Estimation of land surface evaporation using a generalized nonlinear 352 complementary relationship. J. Geophys. Res. Atmos., 122(3), 1475-1487, 2017. 353 https://doi.org/10.1002/2016jd025936 354 Zhou, H., Han, S., & Liu, W.: Evaluation of two generalized complementary functions for annual 355 evaporation estimation on the Loess Plateau, China. J. Hydrol., 124980, 2020. 356 357 https://doi.org/10.1016/j.jhydrol.2020.124980

358

**360 Appendix:**

361

**SPRINGER NATURE Author Services**

**Editing Certificate**

This document certifies that the manuscript

At which time scale does the complementary principle perform best on evaporation estimation?

prepared by the authors

**Liming Wang, Songjun Han, Fuqiang Tian**

was edited for proper English language, grammar, punctuation, spelling, and overall style by one or more of the highly qualified native English speaking editors at SNAS.

This certificate was issued on **October 26, 2020** and may be verified on the SNAS website using the verification code **38A9-7806-57F3-27BD-66AB**.

Neither the research content nor the authors' intentions were altered in any way during the editing process. Documents receiving this certification should be English-ready for publication; however, the author has the ability to accept or reject our suggestions and changes. To verify the final SNAS edited version, please visit our verification page at secure.authorservices.springernature.com/certificate/verify.
If you have any questions or concerns about this edited document, please contact SNAS at support@as.springernature.com.

SNAS provides a range of editing, translation, and manuscript services for researchers and publishers around the world. For more information about our company, services, and partner discounts, please visit authorservices.springernature.com.

| 364 | REAL EVAPOTRANSPIRATION AND POTENTIAL CLIMATIC SIGNIFICANCE                                        |
|-----|----------------------------------------------------------------------------------------------------|
| 365 |                                                                                                    |
| 366 | R.J.BOUCHET                                                                                        |
| 367 | Central station of bioclimatologie, Versailles                                                     |
| 368 | National institute of the Agronomic research (France)                                              |
| 369 |                                                                                                    |
| 370 |                                                                                                    |
| 371 |                                                                                                    |
| 372 | The real evapotranspiration of an area represents the water really lost in the form of vapor,      |
| 373 | the potential evapotranspiration, the water likely to be lost under the same conditions when it    |
| 374 | is not limiting factor any more. The knowledge of these two data is obviously indispensable to     |
| 375 | study the circulation of water or to define the needs for the water of the cultures.               |
| 376 | We propose to show the connections that exist not only between ETP and ETR, but also               |
| 377 | between these terms and the various elements of the energy report (the total radiation, the        |
| 378 | radiation of long wave, etc), by using the method of the energy assessment. The simple             |
| 379 | relations that we will establish will permit to better define the climatic significance of ETR and |
| 380 | ETP. It will then be possible to specify their respective variations when we will try to modify    |
| 381 | the climate in more or less vast zones, either by irrigating, or by changing the cover of the      |
| 382 | ground.                                                                                            |
| 383 |                                                                                                    |
| 384 |                                                                                                    |
| 385 | I STARTING ASSUMPTION SCALE OF THE ASSESSMENT                                                      |
| 386 |                                                                                                    |
| 387 |                                                                                                    |
| 388 | The study of the energy assessment supposes the preliminary definition of the system               |
| 389 | limits. To avoid taking into account the phenomena of accumulation and restitution of heating      |
| 390 | during the diurnal and night phases, the assessment will relate to one 24-hour period, the         |
| 391 | variations of temperature are then generally negligible.                                           |
| 392 | The system includes the whole of the vegetable mass, a superficial section of ground, and a        |
| 393 | lower section of the atmosphere. Dimensions of these sections are just as the nycthemeral          |
| 394 | variations of temperature remain appreciable. The system exchange of heating with outside          |
| 395 | during this period takes place without the phenomena of radiation and evaporation, by              |
| 396 | conducting in deep